# Social Media Addiction and Procrastination in Peruvian University Students: Exploring the Role of Emotional Regulation and Age Moderation

**DOI:** 10.3390/healthcare13091072

**Published:** 2025-05-06

**Authors:** Sandra Elizabeth Fuentes Chavez, Velia Graciela Vera-Calmet, Haydee Mercedes Aguilar-Armas, Lucy Angélica Yglesias Alva, Marco Agustín Arbulú Ballesteros, Cristian Edgardo Alegria Silva

**Affiliations:** 1Instituto de Investigación en Ciencias y Tecnología, Universidad César Vallejo, Campus Trujillo, Chepen 13001, Peru; sfuentesc@ucvvirtual.edu.pe (S.E.F.C.); haguilarar@ucvvitual.edu.pe (H.M.A.-A.); marbulub@ucv.edu.pe (M.A.A.B.); calegriasi@ucvvirtual.edu.pe (C.E.A.S.); 2Departamento de Estadística, Facultad de Ciencia Físicas y Matemáticas, Universidad Nacional de Trujillo, Trujillo 13001, Peru; lyglesias@unitru.edu.pe

**Keywords:** emotional regulation, social media addiction, irrational procrastination, irrational procrastination, university students, academic performance, psychological mediation, age moderation, digital behavior, emotional management, academic sustainability

## Abstract

**Objectives**: This study examines the mediating role of emotional regulation in the relationship between social media addiction and irrational procrastination among university students in Trujillo, Peru. **Methods:** The research employed a non-experimental, explanatory design with latent variables using measurement scales involving 342 university students aged 18 to 36 years. Data collection was carried out using quota sampling using institutional email lists. The findings reveal that social media addiction significantly influences both irrational procrastination and emotional regulation, with age moderating the relationship between emotional regulation and procrastination. **Results**: The results indicated that social media addiction explained 9.5% of the variance in procrastination and 12% of the variance in emotional regulation. Interestingly, although age alone did not directly predict procrastination, it demonstrated a significant moderating effect when combined with emotional regulation. The study did not find a significant mediating effect of emotional regulation between social media addiction and procrastination. **Conclusions**: These findings contribute to understanding the complex dynamics between digital behavior, emotional regulation, and academic procrastination, suggesting the need for targeted interventions that consider age-specific approaches to emotional regulation and social media use in the academic setting.

## 1. Introduction

Emotional regulation, social media addiction, and procrastination are interconnected phenomena that significantly impact university students’ academic performance and psychological well-being. This study examines these relationships within the Peruvian university context, where 92.7% of young people under 18 use the internet primarily for entertainment, often prioritizing non-academic activities and postponing educational responsibilities [1,2].

### 1.1. Relationship Between Social Media Addiction and Irrational Procrastination

Social media addiction refers to the compulsive use of online social platforms that interferes with daily functioning. According to Escurra Mayaute and Salas Blas [3], this behavioral addiction manifests through obsession with social networks, lack of personal control, and excessive usage patterns. Irrational procrastination, as defined by Steel [4], is the voluntary postponement of planned activities despite anticipating negative consequences, characterized by irrationality, delay, and dysphoria [5,6].

The relationship between these phenomena is explained by the Temporal Procrastination Model proposed by Steel [4], which posits that procrastination results from seeking immediate gratification while avoiding emotional discomfort associated with challenging tasks. Social networks provide an accessible escape mechanism, offering immediate rewards through notifications, social validation, and entertaining content. Rozgonjuk et al. [7] demonstrated that social media use during lectures mediates the relationship between procrastination and problematic smartphone use, while Tuckman [8] highlighted how the immediacy of rewards interferes with long-term goal achievement.

Sirois and Pychyl [9] further clarified that procrastination involves both positive and negative reinforcement, where social networks temporarily reduce emotional distress while providing instant gratification, strengthening this behavioral pattern. This process can evolve into a cycle of addiction, as Zhuang et al. [10] found, where excessive social media usage becomes a habitual avoidance mechanism, simultaneously increasing both procrastination and platform dependency.

Recent studies by Suárez-Perdomo et al. [11] and Li et al. [12] have confirmed strong associations between social media addiction and academic procrastination among university students, demonstrating how digital behaviors can significantly impair academic performance through the postponement of essential tasks.

### 1.2. The Mediating Role of Emotional Regulation

Emotional regulation—defined by Gross [13] as the ability to implement strategies to transform emotional experiences, externalize them, or respond to environmental demands—plays a crucial mediating role between social media addiction and procrastination. This psychological mechanism encompasses cognitive reappraisal and emotional suppression strategies [14,15,16] that facilitate the adaptation to environmental demands.

The Emotional Regulation and Problematic Internet Use Model developed by Kuss and Griffiths [17] provides a theoretical framework for understanding this relationship. Their model suggests that individuals with emotional regulation difficulties are more likely to use social networks as a secondary emotional management tool, particularly to alleviate negative emotions like anxiety, loneliness, or depression. Griffiths [18] expanded this perspective by exploring how problematic internet use relates to patterns of emotional avoidance and its impact on mental health [19,20,21].

This usage pattern can create a dependency cycle, where users increasingly rely on social platforms for immediate gratification, reducing their ability to manage emotions independently [22]. Furthermore, as Szawloga et al. [23] and Yam et al. [24] observed, this dependency paradoxically increases social isolation in offline contexts, magnifying emotional distress despite the illusion of social connection.

Pychyl and Sirois [25] specifically addressed the influence of emotional regulation on procrastination, positioning it not merely as a time management issue but as a dysfunctional emotion-avoidance mechanism. Their research demonstrated that people procrastinate to avoid negative emotions associated with challenging tasks, opting instead for immediately gratifying activities. Danne et al. [26] supported this by highlighting how emotional self-regulation difficulties correlate with procrastination, while Sirois [27] emphasized that poor emotional regulation skills consistently predict higher procrastination levels, negatively affecting psychological well-being and performance.

### 1.3. The Moderating Role of Age

Age represents a significant moderating factor in the relationship between emotional regulation and procrastination. Steel’s [4] model suggests that younger individuals tend to procrastinate more due to greater impulsivity and emotional regulation difficulties. This pattern received support from Yan and Zhang [28], who emphasized that impulsivity and underdeveloped planning skills—prevalent in younger populations—maximize procrastination tendencies in academic contexts.

As individuals mature, they develop enhanced self-regulation and emotional management skills that typically reduce procrastination behaviors. However, Steel [4] noted that, while procrastination generally decreases with age, it persists in high emotional load or stress situations. Tandon et al. [22] complemented this perspective by identifying self-regulation as a key factor that improves with experience, enabling older individuals to approach tasks more efficiently with less avoidance behavior.

The Socioemotional Selectivity Theory proposed by Carstensen [29] offers additional insights, suggesting that emotional priorities and regulation abilities change with age, influencing procrastination tendencies. This model argues that younger individuals focus on long-term goals such as knowledge acquisition and personal development, leading them to prioritize avoiding immediate discomfort associated with challenging tasks. Consequently, they exhibit higher procrastination rates compared to adults who have developed superior emotional management capabilities.

As people age, Sirois [27] observed that they tend to prioritize emotional regulation and immediate well-being, reducing stress associated with incomplete tasks. Griffiths [18] further noted that older individuals demonstrate lower procrastination levels due to more effective management of negative emotions like anxiety and stress, enabling more direct task engagement. These age-related differences highlight the importance of developing appropriate emotional regulation skills throughout life to mitigate irrational procrastination effects.

### 1.4. Research Objectives and Hypotheses

Therefore, the objective of this research is to evaluate the mediating role of emotional regulation in the relationship between social media addiction and irrational procrastination in university students while also examining the moderating effect of age on these relationships. The study seeks to provide valuable insights for developing strategies and interventions that enhance emotional regulation to improve academic performance and emotional well-being among university students, see Table 1.

Based on the literature, the following hypotheses are proposed:

**H1.** 
*Social media addiction has a significant effect on irrational procrastination [4,7,8,9,10].*


**H2.** 
*Addiction to social networks has a significant effect on emotional regulation [17,27,28].*


**H3.** 
*Age has a significant effect on irrational procrastination [4,22,26,28].*


**H4.** 
*Emotional regulation has a significant effect on irrational procrastination [33,34,35].*


**H5.** 
*Age moderates the relationship between emotional regulation and irrational procrastination, such that the influence of emotional regulation on irrational procrastination varies depending on the student’s age [4,28,29,36].*


**H6.** 
*Social media addiction has a significant indirect effect on irrational procrastination, as mediated by emotional regulation [4,30,37].*


### 1.5. Importance of the Study

This research addresses a growing concern in university settings regarding the increasing prevalence of social media addiction and procrastination behaviors, which significantly impact both academic performance and emotional well-being. University students face numerous academic and social responsibilities, making it crucial to understand how emotional regulation influences these problematic behaviors.

The findings will contribute to university students’ professional development by identifying factors affecting academic performance, enabling the implementation of performance-enhancing strategies. Additionally, this knowledge provides valuable foundations for higher education institutions to design targeted interventions that foster positive attitudes and skills, promoting improved academic performance during professional training [38,39].

The results obtained will be fundamental for the future professional development of university students, as, by identifying the factors that affect their academic performance, they will be able to implement strategies that improve their performance in the professional sphere. Furthermore, this knowledge provides a valuable basis for higher education institutions in designing and implementing interventions aimed at fostering positive attitudes and skills, promoting better academic performance during professional training [38,39].

## 2. Materials and Methods

### 2.1. Approach and Design

This study assesses the moderating effect of age on the relationship between social media addiction and irrational procrastination, as well as the mediating role of emotional regulation as shown in Figure 1 [40].

Since the analysis was performed with SmartPLS 4.0, which uses bootstrapping for significance testing, a normality test was not required. Bootstrapping is a resampling technique that does not assume normality in the data and provides robust results even with non-normal distributions.

### 2.2. Population and Sample

A total of 342 university students from Trujillo-Peru participated. The participants ranged in age from 18 to 36 years old. Data collection was carried out from January to March 2024 through quota sampling. After establishing the quotas for gender and age, participants were randomly selected from the list of student institutional email addresses, ensuring that each quota was filled with a random sample of the corresponding student population. Participants were recruited through a list of student institutional email addresses, and invitations were sent until the quotas for each group were filled. The use of institutional email lists was handled in accordance with ethical guidelines and approved by the Ethics Committee 2024-IIICyT-ITCA (Approval Code: 0120-2024-GM-IIICyT-IIICyT). Prior to accessing these lists, proper authorization was obtained from the university authorities. Students’ privacy was protected through several measures: (1) emails were sent using blind carbon copy (BCC) to prevent disclosure of recipients, (2) participation was entirely voluntary, with no academic incentives or penalties, (3) participants could opt out at any time, and (4) all data were anonymized during analysis. All email addresses were deleted from the research database once data collection was completed, with only the anonymized responses retained for analysis.

### 2.3. Ethical Considerations

This study was approved by the Ethics Committee 2024-IIICyT-ITCA (Approval Code: 0120-2024-GM-IIICyT-IIICyT-IIICyT, Approval Date: 18 March 2024). The use of institutional email lists for contacting participants was authorized by the corresponding university authorities. All participants were informed about the purpose of the study, the voluntary nature of their participation, and the confidential handling of sensitive data related to the use of social networks and procrastination behaviors. Informed consent was obtained from all participants prior to their inclusion in the study, and they were informed of their right to withdraw at any time without negative consequences.

### 2.4. Instruments

The Emotional Regulation Questionnaire (ERQ), which measures how people manage their emotions and was developed by [14], is an instrument composed of 10 items rated on a 7-point Likert scale and distributed in two dimensions: cognitive reappraisal and emotional suppression, translated into Spanish by [41]. The creators of the original scale reported a Cronbach’s alpha reliability of 0.79 and 0.73 for the two dimensions.

The Social Media Addiction Questionnaire (ARS), developed by [3], contains 24 items, each of which is scored on a 5-point Likert scale (1 = never, 2 = almost never, 3 = sometimes, 4 = almost always, and 5 = always), with a three-dimensional structure: obsession with social networks, lack of personal control in the use of social networks, and excessive use of social networks, with respective Cronbach’s alpha coefficients of 0.91, 0.88, and 0.92.

The Irrational Procrastination Questionnaire (IPS), adapted for Mexican university students with an age range of 18 to 56 years, was used [42]. The instrument has seven elements with five possible choices; its unidimensional structure was corroborated by exploratory and confirmatory factor analysis with adequate fit indicators (greater than 0.90) and an RMSEA of 0.045, obtaining a Cronbach’s alpha coefficient of 0.80 and structural validity.

### 2.5. Procedure and Statistical Analysis

An exploratory factor analysis (EFA) was performed using IBM SPSS 26, incorporating all items from the three instruments. Factorization was performed using the principal axis method with Promax oblique rotation. To ensure the adequacy of the analysis, Bartlett’s sphericity test was applied, which confirmed the correlation between variables with a *p*-value of less than 0.05, supporting the relevance of the procedure. Similarly, the Kaiser–Meyer–Olkin Index (KMO) was used to assess the adequacy of the sample, indicating acceptable values. Following Ref. [43], items with factor weights below 0.39 and communalities below 0.30 were removed. On the basis of these results, a model was developed that included dimensionality, factor loadings, and explained the variance. Finally, to examine the relationship between latent and observable variables, a structural equation modeling (SEM) analysis was performed using SmartPLS 4.0 software. Since the analysis was performed with SmartPLS 4.0, which uses bootstrapping for significance testing, a normality test was not required. Bootstrapping is a resampling technique that does not assume normality in the data and provides robust results even with non-normal distributions.

### 2.6. Construct Quality Tests

When examining the quality and validity criteria of the study, the data painted a complex and nuanced picture of the robustness of the research model. For example, the social media addiction scale has a Cronbach’s alpha coefficient of 0.954, which means that it is very reliable. The procrastination and emotional regulation scales also have good values, 0.850 and 0.865, respectively, which means that the measures are very consistent with each other. Convergent validity, reflected in the average extracted variance (AVE), shows values above 0.5 for all the scales (0.502 for social media addiction, 0.570 for procrastination, and 0.583 for emotional regulation), suggesting that the items of each scale effectively measure the constructs they are intended to measure and thus contribute to the overall robustness of the measurement model (see Table 2).

The results demonstrate the discriminant validity between the main variables in the causal model (see Table 3). The causal pathway from social media addiction to emotional regulation shows a particularly strong effect (0.322), consistent with hypothesis H2. Similarly, the effect of social media addiction on procrastination (0.259) supports hypothesis H1. These findings validate the distinct causal relationships proposed in our SEM model while maintaining their conceptual independence.

## 3. Results

Table 4 shows information on the sociodemographic characteristics of university students, including data such as gender, semester of studies, age, and place of origin. The data show a fascinating reality in Peruvian university students today: women are a clear majority, representing 80% of the student body. This is not coincidence but reflects how certain careers, especially those focused on human and social development, have been attracting more and more female talent in recent decades. If we look at the semester of studies, we find interesting results, such as the fact that the tenth cycle concentrates on more than 26% of the students, which speaks of a group that has been able to adapt and persist despite all the changes that higher education has faced in recent times. They are students who have shown great capacity for adaptation, moving between face-to-face and virtual classes without losing their way. The age of the students tells another part of the story, as half of them are between 18 and 22 years old, who have grown up with technology as a natural part of their lives and are living their university education at a time of great social change. But there is also a valuable group, around 15%, who are over 28 years old and bring with them work and life experiences that enrich class discussions and group work. The origin of the students shows interesting data, as many of them are from the Peruvian north, with Trujillo being the main center, contributing almost half of the students, but the significant presence of young people from Cajamarca (12.58%) and Huamachuco (6.44%) shows that the university has become an important regional educational center. This has created natural support networks between students from different cities, strengthening the role of the institution in the region.

The descriptive results reveal a complex picture in the relationship between social media addiction, emotional regulation, and procrastination. While half of the students show medium levels of social media addiction, 10.2% have high levels, suggesting problematic use that could affect their well-being and academic performance. Regarding emotional regulation, emotional suppression seems to be the most used strategy, with 50% of the students presenting medium levels and 25.7% high levels. This could indicate difficulties in managing emotions adaptively, which, in turn, could influence procrastination. Finally, 22.2% of the students show high levels of procrastination, suggesting the need for interventions that promote time management and emotional regulation.

Table 5 presents the R-squared and adjusted R-squared values for the variables procrastination and emotional regulation. The results indicate that the model explains 9.5% of the variability of procrastination and 12% of emotional regulation, with smaller adjustments after taking into account the penalty for the number of predictors in the model. These values reflect a moderate ability of the model to predict both variables.

The multicollinearity analysis, assessed by the Variance Inflation Factor (VIF), shows generally acceptable values below 5.0, with most items ranging between 1.2 and 4.0, and the ADIRED7 item shows the highest value (4.036), although still within tolerable limits, suggesting that there are no significant redundancy problems between the predictor variables (see Table 6).

In Table 7, the fit indices of the model show interesting aspects, as while the SRMR of 0.076 is within the acceptable range, the NFI of 0.712 suggests a moderate fit of the model to the data, while the R-squared for procrastination (0.095) and emotional regulation (0.120) indicate that, although there is a predictive effect, there are other factors not considered in the model that could be influencing these variables. The consolidated results are shown in Figure 2.

Table 8 provides an interesting snapshot of the relationship between social networks, emotions, and the tendency to procrastinate among university students. The study demonstrates a causal relationship wherein social media addiction significantly influences irrational procrastination among students, supporting H1. Additionally, consistent with H2, excessive social media use was found to negatively impact students’ emotional regulation capabilities, establishing a clear directional effect. These causal relationships, validated through our SEM analysis, provide strong support for the theoretical framework proposing that social media addiction functions as an antecedent variable that affects both emotional regulation and procrastination behaviors. However, something curious emerged in the results: age alone does not determine whether someone will procrastinate, nor does being good or bad at managing emotions directly predict whether someone will procrastinate. What is interesting is how age and emotion management interact. The results suggest that the way students manage their emotions affects their tendency to procrastinate differently depending on their age. It is as if the maturity acquired over the years changes the way emotions influence our decisions to do or postpone tasks. An unexpected finding was that, although excessive use of social networks affects both emotions and procrastination, there was not as direct a link between these elements as previously thought. In other words, it is not as simple as saying that social networks affect emotions and that is why students procrastinate. All this leads us to think that, when you want to help students better manage their time and reduce procrastination, it is not enough to tell them to use social networks less. You also need to consider how they manage their emotions and take into account that different age groups may need different types of support and strategies.

## 4. Discussion

### 4.1. The Relationship Between Social Media Addiction and Procrastination

Several studies have shown a significant correlation between social media addiction and procrastination, particularly in academic contexts, as demonstrated by this study. For instance, Ref. [44] found that excessive use of social networks is associated with increased procrastination in academic activities, while Ref. [33] identified a positive relationship between the risk of addiction to these platforms and procrastination in students. Similarly, Ref. [45] noted that removing the Facebook newsfeed helped users avoid distractions and improve focus, indicating that certain design elements on social networks encourage procrastination behaviors. These findings highlight the urgency of implementing strategies that enable more controlled use of social networks, minimizing their adverse impact on university students’ academic performance and productivity [46,47].

### 4.2. The Role of Emotional Regulation in Procrastination

Although emotional regulation was hypothesized to mediate the relationship between social media addiction and irrational procrastination, the results did not reveal a significant mediating effect. One possible explanation for this could be the complexity of emotional regulation itself, which may interact with other underlying factors that were not considered in this study. Future research could explore additional moderating variables, such as self-esteem, task importance, or coping mechanisms, to provide a more nuanced understanding of how emotional regulation interacts with social media addiction and procrastination. Incorporating these variables could lead to a better identification of the precise pathways through which social media addiction influences procrastination behaviors.

Moreover, the relationship between social media addiction and emotional regulation has been widely studied, revealing a significant connection. Ref. [48] found that excessive use of these platforms negatively impacts students’ emotional regulation, making it harder to assess and manage emotions. Similarly, Ref. [49] identified that problematic use of social networks in adolescents has a detrimental effect on mental health and emotional regulation. Ref. [50] emphasized that addictive patterns in social media use are associated with greater emotional impulsivity in young university students, underscoring the need to develop strategies to balance time spent on social networks and strengthen emotional regulation skills. These findings highlight the importance of targeted interventions to mitigate the negative effects of excessive social media use.

### 4.3. The Impact of Age on Procrastination

The relationship between age and procrastination varies across contexts. In the present study, age and procrastination did not show a significant correlation, which contrasts with [51], who observed that academic procrastination tends to decrease with age. This suggests that older university students procrastinate less compared to their younger counterparts. Similarly, Ref. [52] concluded that procrastination is more prevalent among younger individuals and tends to decrease with age. These findings imply that the development of self-regulation and time management skills over the years could explain the reduction in procrastination in older individuals or that procrastination is used as a coping mechanism to alleviate stress [53,54,55].

### 4.4. Emotional Regulation and Procrastination

In this study, emotional regulation and procrastination were not significantly correlated. However, research by Ref. [56] found that difficulties in managing negative emotions are closely associated with procrastination, which can lead to the delay of important tasks. Ref. [57] noted that poor emotional management is positively related to increased procrastination and negative affectivity, while adequate positive affectivity may reduce procrastination. Refs. [58,59,60,61] demonstrated that strengthening emotional regulation skills can help reduce procrastination behaviors, highlighting the importance of interventions aimed at improving emotional regulation. It is plausible to infer that university students may exhibit both variables, but they are not necessarily connected.

### 4.5. The Moderating Influence of Age on Emotional Regulation and Procrastination

Age could moderate the relationship between emotional regulation and procrastination. Ref. [62] identified that both emotional regulation and academic performance are significant predictors of academic procrastination in university students. Ref. [63] examined the connection between academic stress, procrastination, and psychological well-being in undergraduate students, concluding that procrastination does not act as a moderator between academic stress and psychological well-being, although it does affect the relationship between stressors and the development of symptoms associated with academic stress. These results highlight that, although there is a relationship between emotional regulation and procrastination, factors such as age and context can shape this connection, underscoring the relevance of considering additional variables when addressing procrastination across different student populations. [64]

### 4.6. Social Media Addiction as a Key Mediator

Social media addiction has been identified as a key mediator between emotional regulation and procrastination, underscoring the complex interaction between these variables. Ref. [65] found that increased use of social networks is directly related to heightened academic procrastination, suggesting that excessive time spent on these platforms contributes to procrastination in academic activities. Likewise, Ref. [66] analyzed the impact of problematic social media use on emotional regulation, concluding that such emotional difficulties may intensify procrastination behaviors. Ref. [67] highlighted that social media addiction acts as a significant cause of irrational procrastination, emphasizing its role in mediating between poor emotional regulation and procrastination. These studies suggest that social media addiction can exacerbate emotional difficulties, which, in turn, foster procrastination. This underscores the need to design interventions that reduce the negative impact of social media, promote healthy use, and strengthen emotional regulation skills to improve academic performance.

### 4.7. The Role of Contextual Variables

It is important to recognize that contextual variables, such as socioeconomic background and family support, can significantly influence college students’ procrastination behaviors. Although this study primarily focused on the interactions between social media addiction and emotional regulation, it did not explore how external factors—such as students’ socioeconomic conditions and family support—might moderate these relationships. Students facing economic challenges or lacking strong family support may experience higher levels of academic stress, which could exacerbate procrastination as a coping mechanism in response to anxiety or emotional distress.

Future studies should examine how these contextual variables interact with internal factors (such as emotional regulation and social media addiction) to shape procrastination behaviors. Including measures of family support, social networks, and socioeconomic conditions could provide valuable insights into the most effective intervention strategies. By integrating these factors, a more comprehensive approach could be developed that addresses not only individual factors but also the environmental influences contributing to procrastination.

### 4.8. The Impact of Social Media Content

One crucial aspect that must be considered is the variability in the types of content consumed on social networks. While this study has generally examined social media addiction, it has not distinguished between different types of content (e.g., educational vs. recreational) that students consume, which might influence their emotions and procrastinatory behaviors in different ways. The nature of the social media content could have a significant impact on how students manage their emotions, which, in turn, affects their procrastination. Highly rewarding, immediate content—such as real-time social interactions or the consumption of entertaining material—may encourage procrastination by providing an emotional distraction that reduces immediate stress but increases long-term anxiety due to unfinished academic tasks [60,68,69,70,71].

Therefore, it is recommended that future studies investigate in greater detail how different categories of content (e.g., educational, professional, or entertainment) differentially influence emotional regulation and procrastination. Analyzing content preferences could provide a clearer understanding of the underlying mechanisms linking social media use to academic procrastination. Additionally, interventions aiming to reduce procrastination should not only address the amount of time spent on social networks but also the type of content consumed. Encouraging a more balanced and mindful use of social media that favors emotional regulation may be key to mitigating procrastination [72,73].

### 4.9. Heterogeneity of Internet Users Based on Personality Traits

A critical limitation of the current study is treating internet users as a homogeneous group without considering underlying personality differences that might influence both social media use patterns and procrastination tendencies. Rather than viewing university students as a uniform population, future research would benefit significantly from incorporating personality typologies, particularly those based on the Big Five personality traits (openness, conscientiousness, extraversion, agreeableness, and neuroticism), into the analysis framework.

The Big Five personality traits have been consistently associated with different patterns of internet usage and procrastination behaviors. For instance, individuals high in neuroticism might use social media as an emotional regulation tool, while those high in conscientiousness typically demonstrate lower procrastination tendencies regardless of social media exposure. Extraversion may influence the types of social media content sought and the gratification obtained from online interactions, potentially affecting the addictive potential of these platforms.

Future studies should consider employing personality assessments alongside measures of social media addiction and procrastination to develop more nuanced models that account for individual differences. This approach would enable researchers to identify which personality types are more vulnerable to the negative effects of social media on emotional regulation and procrastination, potentially leading to more targeted interventions. Additionally, examining how personality traits moderate the relationships between the key variables in our model could explain some of the variance not accounted for in the current analysis.

Incorporating personality variables would also enhance the practical applications of this research, as universities could develop more personalized strategies to address procrastination that account for different student personality profiles rather than implementing one-size-fits-all approaches.

## 5. Conclusions

This study explores the relationship between social media addiction, emotional regulation, and procrastination in university students’ lives. The findings indicate a direct correlation between social media addiction and procrastination, with younger students demonstrating a particularly pronounced association. This finding aligns with the observations of other researchers, who have also noted that excessive social media use impedes students’ emotional regulation and hinders their ability to focus on academic tasks. Consequently, interventions aimed at curbing social media use and fostering emotional regulation skills in students may yield positive outcomes, including enhanced academic performance. Furthermore, our findings indicate that age plays a significant role in the manifestation of procrastination among students. As students progress through their education, they tend to exhibit greater self-control and time management skills, which contribute to reduced procrastination. Conversely, younger students may utilize procrastination as a coping mechanism for stress. Consequently, educational institutions should devise distinct strategies for different age groups to assist students in overcoming procrastination. While the present study did not establish a direct correlation between emotional regulation and procrastination, extant research points to the potential of enhanced emotional skills in mitigating procrastination. In light of this, it is recommended that universities consider the development of customized programs aimed at assisting students in managing emotional challenges, given their documented impact on academic performance. The cultivation of emotional intelligence and the integration of constructive coping mechanisms could facilitate students’ navigation of procrastination and enhance their overall well-being.

It is crucial to acknowledge the significant impact of both age and environmental factors on the relationship between emotions and procrastination. This underscores the necessity for a more nuanced understanding of the mechanisms underlying procrastination across diverse student populations. In the development of interventions, it is imperative to consider not only students’ emotional factors but also their background, family support, and social context.

The present study demonstrates that social media addiction exacerbates emotional difficulties, thereby engendering a cycle in which students encounter challenges in effectively managing both their emotions and their academic responsibilities. To effectively address this issue, strategies must encompass a reduction in social media usage in conjunction with a heightened awareness of students’ media consumption. The integration of balanced social media utilization with the cultivation of emotional skills has the potential to serve as a pivotal strategy in the mitigation of procrastination and the enhancement of academic success.

### Limitations of the Study

Despite the valuable insights provided by this research, several limitations should be acknowledged. First, the cross-sectional design of this study prevents establishing definitive causal relationships between the variables, despite the use of structural equation modeling. Longitudinal studies would be necessary to confirm the proposed directional effects and examine how these relationships evolve over time, particularly throughout students’ academic careers.

Second, the sample, although adequate in size, was predominantly female (80.06%) and concentrated on certain semesters of studies, which may have limited the generalizability of the findings. Future research should aim for more balanced gender representation and broader distribution across academic levels to enhance the external validity.

Third, the reliance on self-report measures for all variables introduces the possibility of common method bias and social desirability effects, particularly when assessing sensitive behaviors such as social media addiction and procrastination. Incorporating objective measures of social media usage (such as screen time data) and academic procrastination (such as submission timestamps for academic assignments) could provide more reliable indicators.

Fourth, while the study explains 9.5% of the variance in procrastination and 12% of the emotional regulation, the moderate R-squared values suggest that other significant factors not included in the model may play important roles in these phenomena. As discussed previously, personality traits, specific types of social media content, and contextual variables such as socioeconomic status and family support are likely to account for the additional variance.

Finally, the research was conducted at a specific point in time within a particular cultural context (Peruvian university students), and the findings may not directly translate to other cultural settings or age groups. Cross-cultural studies would help determine which aspects of the observed relationships are universal and which are culturally specific.

These limitations provide a valuable direction for future research while contextualizing the contributions of the current study to our understanding of the complex relationships between social media addiction, emotional regulation, and academic procrastination.

## Figures and Tables

**Figure 1 healthcare-13-01072-f001:**
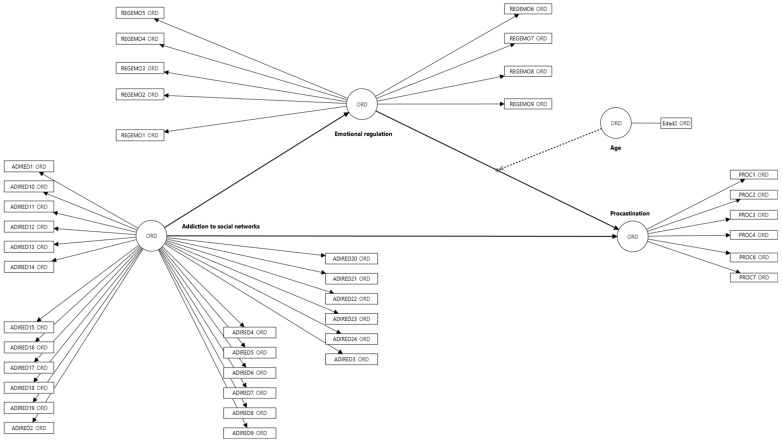
Proposed model.

**Figure 2 healthcare-13-01072-f002:**
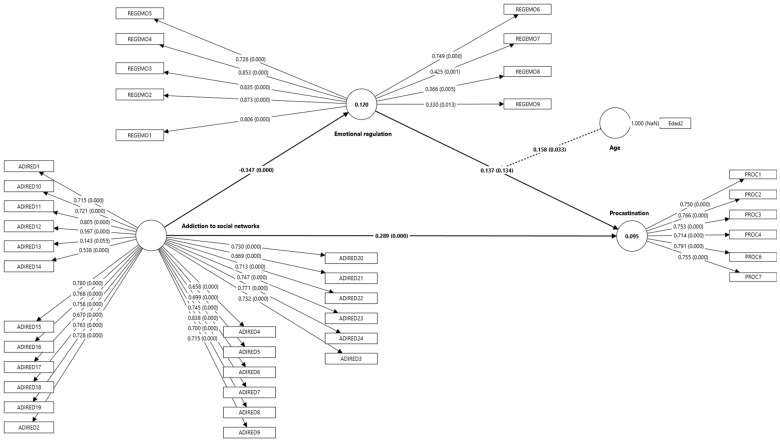
Resolved conceptual research model.

**Table 1 healthcare-13-01072-t001:** Justification of the research hypotheses.

Relation	Author	Explanation
Social Media Addiction—Irrational Procrastination	[4]	The model provides a comprehensive explanation of irrational procrastination as the search for immediate gratification and the escape from emotional discomfort, using social networks as a way to fill emotional gaps, making it difficult to complete important tasks.
Social media addiction: emotional regulation	[17]	The model illustrates how the relationship between social media addiction and emotional regulation is interconnected, with the compulsive use of social media to regulate emotions, reinforcing addictive behavior.
Age—Irrational Procrastination	[4]	The Temporal Procrastination Model explains that as people age, their perception of time and their ability to regulate motivation change, leading to a decrease in procrastination in many cases.
Emotional Regulation—Irrational Procrastination	[25]	The model highlights how procrastination is used as a mechanism to regulate emotions, especially in the context of anxiety and stress.
Age × Emotional regulation: Irrational Procrastination	[29]	The Selective Socioemotional Development Model suggests that as people age, their priorities and motivations change, which affects how they regulate emotions and how they manage procrastination.
Social media addiction—emotional regulation—irrational procrastination	[4,30,31]	Information Processing and Self-Regulation Model, combined with the Socio-Emotional Development Theory and the Procrastination Theory perspective. It stresses that these variables are particularly interrelated, especially in the early stages of life, when self-regulation skills are not fully developed.
Age—Social Media Addictionvan	[32]	Age is related to addictive smartphone behavior, including problematic social media use.

**Table 2 healthcare-13-01072-t002:** Reliability and validity of the constructs.

	Cronbach’s Alpha	Composite Reliability (rho_a)	Composite Reliability (rho_c)	Average Variance Extracted (AVE)
Network addiction	0.954	0.963	0.959	0.502
Procrastination	0.850	0.853	0.888	0.570
Emotional regulation	0.865	0.923	0.884	0.583

**Table 3 healthcare-13-01072-t003:** Discriminant validity—Heterotrait–Monotrait Ratio Matrix (HTMT).

	Network Addiction	Age	Procrastination	Emotional Regulation	Age × Emotional Regulation
Network addiction					
Age	0.136				
Procrastination	0.259	0.034			
Emotional regulation	0.322	0.067	0.159		
Age × Emotional regulation	0.086	0.087	0.163	0.124	

**Table 4 healthcare-13-01072-t004:** Summary of the sociodemographic information.

**Variable**	**Category**	**Absolute Frequency**	**Percentage**
Sex	Female	261	80.06%
	Male	65	19.94%
Cycle	10	85	26.07%
	9	45	13.80%
	8	45	13.80%
	11	44	13.50%
	7	43	13.19%
	6	27	8.28%
	2	17	5.21%
	4	6	1.84%
	5	5	1.53%
Age	18–22 years old	164	50.31%
	23–27 years old	113	34.66%
	28–32 years of age	24	7.36%
	33–37 years	15	4.60%
	38 years and older	10	3.07%
Place of origin	Trujillo	157	48.16%
	Cajamarca	41	12.58%
	Huamachuco	21	6.44%
	Lima	8	2.45%
	Other	29	2.15%
Descriptive results			
**Variable**	**Category**	**Frequency**	**Percentage**
Cognitive Reappraisal	Low	103	30.10%
	Medium	156	45.60%
	High	83	24.30%
Emotional Suppression	Low	83	24.30%
	Medium	171	50.00%
	High	88	25.70%
Social Media Addiction	Low	136	39.80%
	Medium	171	50.00%
	High	35	10.20%
Procrastination	Low	95	27.80%
	Medium	171	50.00%
	High	76	22.20%

**Table 5 healthcare-13-01072-t005:** Regression between procrastination, emotional regulation, and social media addiction.

	R Square	Adjusted R-Squared	F Square
Procrastination	0.095	0.084	0.137
Emotional regulation	0.120	0.117	0.079

Note. Path coefficients are on the left side of the intersections, and *p*-values are on the right side (in brackets).

**Table 6 healthcare-13-01072-t006:** Multicollinearity analysis.

Variable	Range VIF	Media VIF	Critical Elements (VIF > 3.0)
Internet Addiction (ADIRED)	1.238–4.036	2.627	ADIRED7 (4.036), ADIRED19 (3.195), ADIRED11 (3.107), ADIRED10 (3.085), ADIRED6 (3.000)
Procrastination (PROC)	1.515–2.464	1.868	None
Emotional regulation (REGEMO)	1.863–3.238	2.41	REGEMO4 (3.238)

**Table 7 healthcare-13-01072-t007:** Goodness of fit test.

	Saturated Model	Estimated Model
SRMR	0.076	0.076
d_ULS	4.680	4.708
d_G	1.453	1.454
Chi-squared	2481.036	2482.803
NFI	0.713	0.712

**Table 8 healthcare-13-01072-t008:** Hypothesis testing.

Hypothesis	Type of Hypothesis.	Original Sample (O)	Standard Deviation	Statistics t	*p*-Values	Decision
Social media addiction -> Procrastination	Direct	0.289	0.067	4.275	0.000	Reject the null hypothesis
Social media addiction -> Emotional regulation	Direct	−0.347	0.068	5.068	0.000	Reject the null hypothesis
Age -> Procrastination	Direct	0.006	0.058	0.111	0.912	Maintain the null hypothesis
Emotional regulation -> Procrastination	Direct	0.137	0.091	1.498	0.134	Maintain the null hypothesis
Age × Emotional regulation -> Procrastination	Moderation	0.158	0.074	2.139	0.033	Reject the null hypothesis
Social media addiction -> emotional regulation -> procrastination	Mediation	−0.047	0.035	1.347	0.178	Maintain the null hypothesis
Addiction to social networks -> Procastination	Total	0.241	0.062	4.175	0.000	Reject the null hypothesis

## Data Availability

The data are contained within the article.

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
