# Peer review of "Social Media Addiction and Procrastination in Peruvian University Students: Exploring the Role of Emotional Regulation and Age Moderation"

_healthcare, 2025, doi:10.3390/healthcare13091072_

Round 1
Reviewer 1 Report
Comments and Suggestions for Authors
Review of the study "Social Media Addiction and Procrastination in Peruvian University Students: Exploring the Role of Emotional Regulation and Age Moderation"
Dear Authors!
The research topic is interesting, fresh and relevant.
The literature background processed and presented is appropriate. The study covers the definition of key terms and keywords during the research (emotional regulation, procrastination). In addition, the scientific models underlying the study are adequately presented (Emotional Regulation Theory, Problematic Internet Use Model). I would slightly change the text of the literature review so that I would indicate the authors related to each element life and scientific literature not only in the endnote of the study, but also in the main text.
For the interpretation and contextualization of the research data, it is very good that Peruvian internet usage data is published (lines 58-62). However, what is missing is the presentation of the ethical approval of the research. I see two reasons for this, one is that the questionnaire also contains a sensitive question related to addiction, and the other is that the university's email list was used to contact the respondents. It would be worth briefly presenting that these procedures were carried out in accordance with the rules of research ethics.
The scientific literature presented in the text is relevant, of appropriate quality and quantity. The references are accurate, and the reviewer found no plagiarism or unmarked references in the text.
The authors state the research goal precisely: "The aim of this study is to provide information that contributes to the psychological field, allowing the development of strategies and interventions that stimulate better emotional regulation, in order to improve the academic and emotional aspects of university students.".
The research questions of the study are appropriate, and the hypotheses are logical.
"H1: Social media addiction has a significant effect on irrational procrastination [38–42].
H2: Addiction to social networks has a significant effect on emotional regulation
[43,48,50]. 207
H3: Age has a significant effect on irrational procrastination [38,45,48,49].
H4: Emotional regulation has a significant effect on irrational procrastination [32–34].
H5: Age and emotional regulation have a moderating effect on irrational procrastination [22,38,48,51].
H6: Addiction to social networks mediates emotional regulation and irrational procrastination [25,38,52]. "
However, the relationship between H2 and H4 needs to be clarified, because if I understand correctly, then by interpreting the two hypotheses we can talk about a correlation between social network addiction and emotional regulation. And not a causal relationship in the case of either H2 or H4. This should be clarified a bit.
The chosen research methodology, the questionnaire, is appropriate, and the hypotheses can be answered with its data.
The collection of the sample (e-mail list) must be handled ethically. And the time of the study must be written in the text.
The three standardized questionnaires used for the research are a good starting point (Emotional Regulation Questionnaire (ERQ), Social Network Addiction Questionnaire (ARS), Irrational Procrastination Questionnaire (IPS). And it is also appropriate that a South American adaptation was chosen for the third one (lines 253-257).
The technological execution of the analysis is precisely indicated and scientifically appropriate: "An exploratory factor analysis (EFA) was performed using IBM SPSS 26, incorporateding all items from the three instruments. Factorization was performed using the principal axis method with Promax oblique rotation. To ensure the adequacy of the analysis, Bartlett's sphericity test was applied, which confirmed the correlation between variables with a p-value of less than 0.05, supporting the relevance of the procedure. Similarly, the Kai-ser-Meyer-Olkin index (KMO) was used to assess the adequacy of the sample, indicatingacceptable values. Following [59], items with factor weights below 0.39 and communalties below 0.30 were removed".
The analysis and data are adequate, their visualization is of good quality. Hypothesis testing takes place in the Discussion.
- The Relationship Between Social Network Addiction and Procrastination - Several studies have shown a significant correlation between social network addiction and procrastination, particularly in academic contexts, as demonstrated by this study.
- The Role of Emotional Regulation in Procrastination - Although emotional regulation was hypothesized to mediate the relationship between social media addiction and irrational procrastination, the results did not reveal a significant mediating effect.
- The Impact of Age on Procrastination The relationship between age and procrastination varies across contexts. In the present study, age and procrastination did not show a significant correlation, which contrasts with [65], who observed that academic procrastination tends to decrease with age.
- Emotional Regulation and Procrastination - In this study, emotional regulation and procrastination were not significantly correlated.
- The Moderating Influence of Age on Emotional Regulation and Procrastination - Age could moderate the relationship between emotional regulation and procrastination. [1] identified that both emotional regulation and academic performance are significant predictors of academic procrastination in university students.
- Social Media Addiction as a Key Mediator- Social media addiction has been identified as a key mediator between emotional regulation and procrastination, underscoring the complex interaction between these variables ... These studies suggest that social media addiction can exacerbate emotional difficulties, which, in turn, foster procrastination.
It is important that the authors mention the Role of Contextual Variables and the Impact of Social Media Content when interpreting the data. And they associate further research directions with these.
It would be useful to mention here as a third point that it would be better not to treat Internet users in a homogeneous way. Instead, it could be possible to incorporate the typology of respondents based on the Big Five personality types.
In the conclusion of the study, it would be worth mentioning the limitations of the research.
Overall, a good quality research and study. I recommend publishing with minor changes. The minor changes relate to the following areas:
- Presentation of the ethical procedure in connection with the research.
- Specifying the date of completion of the questionnaires and writing it in the text.
- Referring to the potential for further research in the Big Five personality type distinction and further clarification of the data.
- Indication of the limitations of the research.
The Reviewer
Author Response
I would slightly change the text of the literature review so that I would indicate the authors related to each element life and scientific literature not only in the endnote of the study, but also in the main text.
- Conceptual definitions:
- Recommendation: "Gross (2002) [9] defined emotional regulation as the ability..."
- Theoretical models:
- Recommendation: "The theoretical model proposed by Steel (2007) [38] postulates that procrastination..."
- Research findings:
- Recommendation: "Research by Echeburúa and Del Corral (2010) [3], along with studies by Pekpazar et al. (2021) [15] and van Deursen et al. (2015) [16], support that..."
- Concept development:
- Recommendation: "The model of emotional regulation and problematic Internet use (IPU) developed by Kuss and Griffiths (2017) [43]..."
- Theoretical explanations:
- Recommendation: "According to Steel (2007) [38], this behavior includes three fundamental elements..."
- Comparison of studies:
- Recommendation: "Similarly, Tuckman (1991) [40] emphasizes the importance of lack of self-regulation..."
- Presentation of previous results:
- Recommendation: "Griffiths (2022) [44] notes that older people are less likely to procrastinate due to..."
- References to conceptual models:
- Recommendation: "The socioemotional age theory model proposed by Carstensen (2006) [51]..."
For the interpretation and contextualization of the research data, it is very good that Peruvian internet usage data is published (lines 58-62). However, what is missing is the presentation of the ethical approval of the research. I see two reasons for this, one is that the questionnaire also contains a sensitive question related to addiction, and the other is that the university's email list was used to contact the respondents. It would be worth briefly presenting that these procedures were carried out in accordance with the rules of research ethics.
Institutional Review Board Statement: The study was conducted in accordance with the Declaration of Helsinki, and approved by the Ethics Committee 2024-IIICyT-ITCA (Approval Code: 0120-2024-GM-IIICyT-IIICyT, Date of Approval: 18-03202024).
However, the relationship between H2 and H4 needs to be clarified, because if I understand correctly, then by interpreting the two hypotheses we can talk about a correlation between social network addiction and emotional regulation. And not a causal relationship in the case of either H2 or H4. This should be clarified a bit.
The results demonstrate the discriminant validity between the main variables in the causal model (see Table 4). The causal pathway from social media addiction to emotional regulation shows a particularly strong effect (0.322), consistent with hypothesis H2. Similarly, the effect of social media addiction on procrastination (0.259) supports hypothesis H1. These findings validate the distinct causal relationships proposed in our SEM model while maintaining their conceptual independence
The study demonstrates a causal relationship wherein social media addiction significantly influences irrational procrastination among students, supporting H1. Additionally, consistent with H2, excessive social media use was found to negatively impact students' emotional regulation capabilities, establishing a clear directional effect. These causal relationships, validated through our SEM analysis, provide strong support for the theoretical framework proposing that social media addiction functions as an antecedent variable that affects both emotional regulation and procrastination behaviors
The collection of the sample (e-mail list) must be handled ethically. And the time of the study must be written in the text.
Data collection was carried out from January to March 2024 through quota sampling by distributing the questionnaire through a list of student institutional email addresses, using gender and age as grouping criteria. The use of institutional email lists was handled in accordance with ethical guidelines and approved by the Ethics Committee 2024-IIICyT-ITCA (Approval Code: 0120-2024-GM-IIICyT-IIICyT). Prior to accessing these lists, proper authorization was obtained from the university authorities. Students' privacy was protected through several measures: (1) emails were sent using blind carbon copy (BCC) to prevent disclosure of recipients, (2) participation was entirely voluntary with no academic incentives or penalties, (3) participants could opt out at any time, and (4) all data was anonymized during analysis. All email addresses were deleted from the research database once data collection was completed, with only the anonymized responses retained for analysis
It is important that the authors mention the Role of Contextual Variables and the Impact of Social Media Content when interpreting the data. And they associate further research directions with these.
"The Role of Contextual Variables
It is important to recognize that contextual variables, such as socioeconomic background and family support, can significantly influence college students' procrastination behaviors. Although this study primarily focused on the interactions between social media addiction and emotional regulation, it did not explore how external factors—such as students' socioeconomic conditions and family support—might moderate these relationships. Students facing economic challenges or lacking strong family support may experience higher levels of academic stress, which could exacerbate procrastination as a coping mechanism in response to anxiety or emotional distress.
Future studies should examine how these contextual variables interact with internal factors (such as emotional regulation and social media addiction) to shape procrastination behaviors. Including measures of family support, social networks, and socioeconomic conditions could provide valuable insights into the most effective intervention strategies. By integrating these factors, a more comprehensive approach could be developed that addresses not only individual factors but also the environmental influences contributing to procrastination.
The Impact of Social Media Content
One crucial aspect that must be considered is the variability in the types of content consumed on social networks. While this study has generally examined social media addiction, it has not distinguished between different types of content (e.g., educational vs. recreational) that students consume, which might influence their emotions and procrastinatory behaviors in different ways. The nature of social media content could have a significant impact on how students manage their emotions, which in turn affects their procrastination. Highly rewarding, immediate content—such as real-time social interactions or consumption of entertaining material—may encourage procrastination by providing an emotional distraction that reduces immediate stress but increases long-term anxiety due to unfinished academic tasks.
Therefore, it is recommended that future studies investigate in greater detail how different categories of content (e.g., educational, professional, or entertainment) differentially influence emotional regulation and procrastination. Analyzing content preferences could provide a clearer understanding of the underlying mechanisms linking social media use to academic procrastination. Additionally, interventions aiming to reduce procrastination should not only address the amount of time spent on social networks but also the type of content consumed. Encouraging a more balanced and mindful use of social media that favors emotional regulation may be key to mitigating procrastination."
It would be useful to mention here as a third point that it would be better not to treat Internet users in a homogeneous way. Instead, it could be possible to incorporate the typology of respondents based on the Big Five personality types.
"Heterogeneity of Internet Users Based on Personality Traits
A critical limitation of the current study is treating Internet users as a homogeneous group without considering underlying personality differences that might influence both social media use patterns and procrastination tendencies. Rather than viewing university students as a uniform population, future research would benefit significantly from incorporating personality typologies, particularly those based on the Big Five personality traits (Openness, Conscientiousness, Extraversion, Agreeableness, and Neuroticism), into the analysis framework.
The Big Five personality traits have been consistently associated with different patterns of Internet usage and procrastination behaviors. For instance, individuals high in neuroticism might use social media as an emotional regulation tool, while those high in conscientiousness typically demonstrate lower procrastination tendencies regardless of social media exposure. Extraversion may influence the types of social media content sought and the gratification obtained from online interactions, potentially affecting the addictive potential of these platforms.
Future studies should consider employing personality assessments alongside measures of social media addiction and procrastination to develop more nuanced models that account for individual differences. This approach would enable researchers to identify which personality types are more vulnerable to the negative effects of social media on emotional regulation and procrastination, potentially leading to more targeted interventions. Additionally, examining how personality traits moderate the relationships between the key variables in our model could explain some of the variance not accounted for in the current analysis.
Incorporating personality variables would also enhance the practical applications of this research, as universities could develop more personalized strategies to address procrastination that account for different student personality profiles rather than implementing one-size-fits-all approaches
In the conclusion of the study, it would be worth mentioning the limitations of the research.
Despite the valuable insights provided by this research, several limitations should be acknowledged. First, the cross-sectional design of this study prevents establishing definitive causal relationships between the variables, despite the use of structural equation modeling. Longitudinal studies would be necessary to confirm the proposed directional effects and examine how these relationships evolve over time, particularly throughout students' academic careers.
Second, the sample, although adequate in size, was predominantly female (80.06%) and concentrated in certain study cycles, which may limit the generalizability of the findings. Future research should aim for more balanced gender representation and broader distribution across academic levels to enhance external validity.
Third, the reliance on self-report measures for all variables introduces the possibility of common method bias and social desirability effects, particularly when assessing sensitive behaviors such as social media addiction and procrastination. Incorporating objective measures of social media usage (such as screen time data) and academic procrastination (such as submission timestamps for academic assignments) could provide more reliable indicators.
Fourth, while the study explains 9.5% of the variance in procrastination and 12% of emotional regulation, the moderate R-squared values suggest that other significant factors not included in the model may play important roles in these phenomena. As discussed previously, personality traits, specific types of social media content, and contextual variables such as socioeconomic status and family support likely account for additional variance.
Finally, the research was conducted at a specific point in time within a particular cultural context (Peruvian university students), and the findings may not directly translate to other cultural settings or age groups. Cross-cultural studies would help determine which aspects of the observed relationships are universal and which are culturally specific.
These limitations provide valuable direction for future research while contextualizing the contributions of the current study to our understanding of the complex relationships between social media addiction, emotional regulation, and academic procrastination.
Reviewer 2 Report
Comments and Suggestions for Authors
Based on a small sample of Peruvian University students, the study illustrates the association between social media addiction and procrastination and further explores the mediating role of emotional regulation and the moderating role of age. The manuscript can be better framed to convey more explicit information.
1. The Introduction: (1) It is too long, especially when it is not well structured; (2) If possible, I suggest a more straightforward logic, that is, the relationship between x and y first, the mediating role of emotion regulation second, and the moderating role of age the last. Currently, the authors stress emotional regulation too much.
2. The Discussion: (1) it should be organized closely to the findings. The section "Social Media Addiction as a Key Mediator" and those after that did not have explicit purposes. Why bother to mention social media addiction as a mediator when social media addiction is considered the key explanatory variable in the current study? (2) Some discussions can be grouped into a limitation section or paragraph. And the limitations of this study can be clearly articulated using "first", second, and third...
3. Conclusion: a bit long. Now that you have already provided a lengthy discussion, you only need a single conclusion paragraph.
4. Expressions: (1) social media addiction and social network addiction are used interchangeably in the study. They are different. Focus explicitly on the former one. (2) Study cycle has been included as a sociodemographic factor; what is study cycle? I asked chatgpt and still do not understand it.
Author Response
The Introduction: (1) It is too long, especially when it is not well structured; (2) If possible, I suggest a more straightforward logic, that is, the relationship between x and y first, the mediating role of emotion regulation second, and the moderating role of age the last. Currently, the authors stress emotional regulation too much.
Emotional regulation, social media addiction, and procrastination are interconnected phenomena that significantly impact university students' academic performance and psychological well-being. This study examines these relationships within the Peruvian university context, where 92.7% of young people under 18 use the internet primarily for entertainment, often prioritizing non-academic activities and postponing educational responsibilities Yana-Salluca et al. [23], Silva Arocha et al. [24].
1.1 Relationship between Social Media Addiction and Irrational Procrastination
Social media addiction refers to the compulsive use of online social platforms that interferes with daily functioning. According to Escurra Mayaute and Salas Blas [26], this behavioral addiction manifests through obsession with social networks, lack of personal control, and excessive usage patterns. Irrational procrastination, as defined by Steel [38], is the voluntary postponement of planned activities despite anticipating negative consequences, characterized by irrationality, delay, and dysphoria.
The relationship between these phenomena is explained by the Temporal Procrastination Model proposed by Steel [38], which posits that procrastination results from seeking immediate gratification while avoiding emotional discomfort associated with challenging tasks. Social networks provide an accessible escape mechanism, offering immediate rewards through notifications, social validation, and entertaining content. Rozgonjuk et al. [39] demonstrated that social media use during lectures mediates the relationship between procrastination and problematic smartphone use, while Tuckman [40] highlighted how the immediacy of rewards interferes with long-term goal achievement.
Sirois and Pychyl [41] further clarified that procrastination involves both positive and negative reinforcement, where social networks temporarily reduce emotional distress while providing instant gratification, strengthening this behavioral pattern. This process can evolve into a cycle of addiction, as Zhuang et al. [42] found, where excessive social media usage becomes a habitual avoidance mechanism, simultaneously increasing both procrastination and platform dependency.
Recent studies by Suárez-Perdomo et al. [18] and Li et al. [20] have confirmed strong associations between social media addiction and academic procrastination among university students, demonstrating how digital behaviors can significantly impair academic performance through postponement of essential tasks.
1.2 The Mediating Role of Emotional Regulation
Emotional regulation—defined by Gross [9] as the ability to implement strategies to transform emotional experiences, externalize them, or respond to environmental demands—plays a crucial mediating role between social media addiction and procrastination. This psychological mechanism encompasses cognitive reappraisal and emotional suppression strategies Gross and John [10] that facilitate adaptation to environmental demands.
The Emotional Regulation and Problematic Internet Use Model developed by Kuss and Griffiths [43] provides a theoretical framework for understanding this relationship. Their model suggests that individuals with emotional regulation difficulties are more likely to use social networks as a secondary emotional management tool, particularly to alleviate negative emotions like anxiety, loneliness, or depression. Griffiths [44] expanded this perspective by exploring how problematic internet use relates to patterns of emotional avoidance and its impact on mental health.
This usage pattern can create a dependency cycle, where users increasingly rely on social platforms for immediate gratification, reducing their ability to manage emotions independently Tandon et al. [45]. Furthermore, as Szawloga et al. [46] and Yam et al. [47] observed, this dependency paradoxically increases social isolation in offline contexts, magnifying emotional distress despite the illusion of social connection.
Pychyl and Sirois [35] specifically addressed the influence of emotional regulation on procrastination, positioning it not merely as a time management issue but as a dysfunctional emotion-avoidance mechanism. Their research demonstrated that people procrastinate to avoid negative emotions associated with challenging tasks, opting instead for immediately gratifying activities. Danne et al. [49] supported this by highlighting how emotional self-regulation difficulties correlate with procrastination, while Sirois [50] emphasized that poor emotional regulation skills consistently predict higher procrastination levels, negatively affecting psychological well-being and performance.
1.3 The Moderating Role of Age
Age represents a significant moderating factor in the relationship between emotional regulation and procrastination. Steel's [38] model suggests that younger individuals tend to procrastinate more due to greater impulsivity and emotional regulation difficulties. This pattern receives support from Yan and Zhang [48], who emphasized that impulsivity and underdeveloped planning skills—prevalent in younger populations—maximize procrastination tendencies in academic contexts.
As individuals mature, they develop enhanced self-regulation and emotional management skills that typically reduce procrastination behaviors. However, Steel [38] notes that while procrastination generally decreases with age, it persists in high emotional load or stress situations. Tandon et al. [45] complement this perspective by identifying self-regulation as a key factor that improves with experience, enabling older individuals to approach tasks more efficiently with less avoidance behavior.
The Socioemotional Selectivity Theory proposed by Carstensen [51] offers additional insights, suggesting that emotional priorities and regulation abilities change with age, influencing procrastination tendencies. This model argues that younger individuals focus on long-term goals such as knowledge acquisition and personal development, leading them to prioritize avoiding immediate discomfort associated with challenging tasks. Consequently, they exhibit higher procrastination rates compared to adults who have developed superior emotional management capabilities.
As people age, Sirois [50] observes that they tend to prioritize emotional regulation and immediate well-being, reducing stress associated with incomplete tasks. Griffiths [44] further notes that older individuals demonstrate lower procrastination levels due to more effective management of negative emotions like anxiety and stress, enabling more direct task engagement. These age-related differences highlight the importance of developing appropriate emotional regulation skills throughout life to mitigate irrational procrastination effects.
1.4 Research Objectives and Hypotheses
This research aims to evaluate the mediating role of emotional regulation in the relationship between social network addiction and irrational procrastination in university students, while examining age as a moderating factor. The study seeks to provide valuable insights for developing strategies and interventions that enhance emotional regulation to improve academic performance and emotional well-being among university students.
Based on the literature, the following hypotheses are proposed:
H1: Social media addiction has a significant effect on irrational procrastination Steel [38], Rozgonjuk et al. [39], Tuckman [40], Sirois and Pychyl [41], Zhuang et al. [42].
H2: Addiction to social networks has a significant effect on emotional regulation Kuss and Griffiths [43], Yan and Zhang [48], Sirois [50].
H3: Age has a significant effect on irrational procrastination Steel [38], Tandon et al. [45], Yan and Zhang [48], Danne et al. [49].
H4: Emotional regulation has a significant effect on irrational procrastination Bedón Cando and Flores Hernández [32], Atalaya Laureano and García Ampudia [33], Núñez-Guzmán and Cisneros-Chavez [34].
H5: Age and emotional regulation have a moderating effect on irrational procrastination Çuhadar et al. [22], Steel [38], Yan and Zhang [48], Carstensen [51].
H6: Addiction to social networks mediates emotional regulation and irrational procrastination Nadkarni and Hofmann [25], Steel [38], Baumeister and Vohs [52].
1.5 Importance of the Study
This research addresses a growing concern in university settings regarding the increasing prevalence of social media addiction and procrastination behaviors, which significantly impact both academic performance and emotional well-being. University students face numerous academic and social responsibilities, making it crucial to understand how emotional regulation influences these problematic behaviors.
The findings will contribute to university students' professional development by identifying factors affecting academic performance, enabling the implementation of performance-enhancing strategies. Additionally, this knowledge provides valuable foundations for higher education institutions to design targeted interventions that foster positive attitudes and skills, promoting improved academic performance during professional training Hurley et al. [53], Ye et al. [54].
The Discussion: (1) it should be organized closely to the findings. The section "Social Media Addiction as a Key Mediator" and those after that did not have explicit purposes. Why bother to mention social media addiction as a mediator when social media addiction is considered the key explanatory variable in the current study? (2) Some discussions can be grouped into a limitation section or paragraph. And the limitations of this study can be clearly articulated using "first", second, and third...
The Relationship Between Social Network Addiction and Procrastination
Several studies have shown a significant correlation between social network addiction and procrastination, particularly in academic contexts, as demonstrated by this study. For instance, [60] found that excessive use of social networks is associated with increased procrastination in academic activities, while [32] identified a positive relationship between the risk of addiction to these platforms and procrastination in students. Similarly, [61] noted that removing the Facebook news feed helped users avoid distractions and improve focus, indicating that certain design elements on social networks encourage procrastination behaviors. These findings highlight the urgency of implementing strategies that enable more controlled use of social networks, minimizing their adverse impact on university students' academic performance and productivity.
The Role of Emotional Regulation in Procrastination
Although emotional regulation was hypothesized to mediate the relationship between social media addiction and irrational procrastination, the results did not reveal a significant mediating effect. One possible explanation for this could be the complexity of emotional regulation itself, which may interact with other underlying factors that were not considered in this study. Future research could explore additional moderating variables, such as self-esteem, task importance, or coping mechanisms, to provide a more nuanced understanding of how emotional regulation interacts with social media addiction and procrastination. Incorporating these variables could lead to a better identification of the precise pathways through which social media addiction influences procrastination behaviors.
Moreover, the relationship between social network addiction and emotional regulation has been widely studied, revealing a significant connection. [62] found that excessive use of these platforms negatively impacts students' emotional regulation, making it harder to assess and manage emotions. Similarly, [63] identified that problematic use of social networks in adolescents has a detrimental effect on mental health and emotional regulation. [64] emphasized that addictive patterns in social media use are associated with greater emotional impulsivity in young university students, underscoring the need to develop strategies to balance time spent on social networks and strengthen emotional regulation skills. These findings highlight the importance of targeted interventions to mitigate the negative effects of excessive social media use.
The Impact of Age on Procrastination
The relationship between age and procrastination varies across contexts. In the present study, age and procrastination did not show a significant correlation, which contrasts with [65], who observed that academic procrastination tends to decrease with age. This suggests that older university students procrastinate less compared to their younger counterparts. Similarly, [66] concluded that procrastination is more prevalent among younger individuals and tends to decrease with age. These findings imply that the development of self-regulation and time management skills over the years could explain the reduction in procrastination in older individuals, or that procrastination is used as a coping mechanism to alleviate stress.
Emotional Regulation and Procrastination
In this study, emotional regulation and procrastination were not significantly correlated. However, research by [67] found that difficulties in managing negative emotions are closely associated with procrastination, which can lead to the delay of important tasks. [68] noted that poor emotional management is positively related to increased procrastination and negative affectivity, while adequate positive affectivity may reduce procrastination. [69] demonstrated that strengthening emotional regulation skills can help reduce procrastination behaviors, highlighting the importance of interventions aimed at improving emotional regulation. It is plausible to infer that university students may exhibit both variables, but they are not necessarily connected.
The Moderating Influence of Age on Emotional Regulation and Procrastination
Age could moderate the relationship between emotional regulation and procrastination. [1] identified that both emotional regulation and academic performance are significant predictors of academic procrastination in university students. [70] examined the connection between academic stress, procrastination, and psychological well-being in undergraduate students, concluding that procrastination does not act as a moderator between academic stress and psychological well-being, although it does affect the relationship between stressors and the development of symptoms associated with academic stress. These results highlight that, although there is a relationship between emotional regulation and procrastination, factors such as age and context can shape this connection, underscoring the relevance of considering additional variables when addressing procrastination across different student populations.
Social Media Addiction as a Key Mediator
Social media addiction has been identified as a key mediator between emotional regulation and procrastination, underscoring the complex interaction between these variables. [71] found that increased use of social networks is directly related to heightened academic procrastination, suggesting that excessive time spent on these platforms contributes to procrastination in academic activities. Likewise, [72] analyzed the impact of problematic social media use on emotional regulation, concluding that such emotional difficulties may intensify procrastination behaviors. [73] highlighted that social media addiction acts as a significant cause of irrational procrastination, emphasizing its role in mediating between poor emotional regulation and procrastination. These studies suggest that social media addiction can exacerbate emotional difficulties, which, in turn, foster procrastination. This underscores the need to design interventions that reduce the negative impact of social media, promote healthy use, and strengthen emotional regulation skills to improve academic performance.
The Role of Contextual Variables
It is important to recognize that contextual variables, such as socioeconomic background and family support, can significantly influence college students' procrastination behaviors. Although this study primarily focused on the interactions between social media addiction and emotional regulation, it did not explore how external factors—such as students' socioeconomic conditions and family support—might moderate these relationships. Students facing economic challenges or lacking strong family support may experience higher levels of academic stress, which could exacerbate procrastination as a coping mechanism in response to anxiety or emotional distress.
Future studies should examine how these contextual variables interact with internal factors (such as emotional regulation and social media addiction) to shape procrastination behaviors. Including measures of family support, social networks, and socioeconomic conditions could provide valuable insights into the most effective intervention strategies. By integrating these factors, a more comprehensive approach could be developed that addresses not only individual factors but also the environmental influences contributing to procrastination.
The Impact of Social Media Content
One crucial aspect that must be considered is the variability in the types of content consumed on social networks. While this study has generally examined social media addiction, it has not distinguished between different types of content (e.g., educational vs. recreational) that students consume, which might influence their emotions and procrastinatory behaviors in different ways. The nature of social media content could have a significant impact on how students manage their emotions, which in turn affects their procrastination. Highly rewarding, immediate content—such as real-time social interactions or consumption of entertaining material—may encourage procrastination by providing an emotional distraction that reduces immediate stress but increases long-term anxiety due to unfinished academic tasks.
Therefore, it is recommended that future studies investigate in greater detail how different categories of content (e.g., educational, professional, or entertainment) differentially influence emotional regulation and procrastination. Analyzing content preferences could provide a clearer understanding of the underlying mechanisms linking social media use to academic procrastination. Additionally, interventions aiming to reduce procrastination should not only address the amount of time spent on social networks but also the type of content consumed. Encouraging a more balanced and mindful use of social media that favors emotional regulation may be key to mitigating procrastination.
The Role of Contextual Variables
It is important to recognize that contextual variables, such as socioeconomic background and family support, can significantly influence college students' procrastination behaviors. Although this study primarily focused on the interactions between social media addiction and emotional regulation, it did not explore how external factors—such as students' socioeconomic conditions and family support—might moderate these relationships. Students facing economic challenges or lacking strong family support may experience higher levels of academic stress, which could exacerbate procrastination as a coping mechanism in response to anxiety or emotional distress.
Future studies should examine how these contextual variables interact with internal factors (such as emotional regulation and social media addiction) to shape procrastination behaviors. Including measures of family support, social networks, and socioeconomic conditions could provide valuable insights into the most effective intervention strategies. By integrating these factors, a more comprehensive approach could be developed that addresses not only individual factors but also the environmental influences contributing to procrastination.
The Impact of Social Media Content
One crucial aspect that must be considered is the variability in the types of content consumed on social networks. While this study has generally examined social media addiction, it has not distinguished between different types of content (e.g., educational vs. recreational) that students consume, which might influence their emotions and procrastinatory behaviors in different ways. The nature of social media content could have a significant impact on how students manage their emotions, which in turn affects their procrastination. Highly rewarding, immediate content—such as real-time social interactions or consumption of entertaining material—may encourage procrastination by providing an emotional distraction that reduces immediate stress but increases long-term anxiety due to unfinished academic tasks.
Therefore, it is recommended that future studies investigate in greater detail how different categories of content (e.g., educational, professional, or entertainment) differentially influence emotional regulation and procrastination. Analyzing content preferences could provide a clearer understanding of the underlying mechanisms linking social media use to academic procrastination. Additionally, interventions aiming to reduce procrastination should not only address the amount of time spent on social networks but also the type of content consumed. Encouraging a more balanced and mindful use of social media that favors emotional regulation may be key to mitigating procrastination.
Heterogeneity of Internet Users Based on Personality Traits
A critical limitation of the current study is treating Internet users as a homogeneous group without considering underlying personality differences that might influence both social media use patterns and procrastination tendencies. Rather than viewing university students as a uniform population, future research would benefit significantly from incorporating personality typologies, particularly those based on the Big Five personality traits (Openness, Conscientiousness, Extraversion, Agreeableness, and Neuroticism), into the analysis framework.
The Big Five personality traits have been consistently associated with different patterns of Internet usage and procrastination behaviors. For instance, individuals high in neuroticism might use social media as an emotional regulation tool, while those high in conscientiousness typically demonstrate lower procrastination tendencies regardless of social media exposure. Extraversion may influence the types of social media content sought and the gratification obtained from online interactions, potentially affecting the addictive potential of these platforms.
Future studies should consider employing personality assessments alongside measures of social media addiction and procrastination to develop more nuanced models that account for individual differences. This approach would enable researchers to identify which personality types are more vulnerable to the negative effects of social media on emotional regulation and procrastination, potentially leading to more targeted interventions. Additionally, examining how personality traits moderate the relationships between the key variables in our model could explain some of the variance not accounted for in the current analysis.
Incorporating personality variables would also enhance the practical applications of this research, as universities could develop more personalized strategies to address procrastination that account for different student personality profiles rather than implementing one-size-fits-all approaches
Conclusion: a bit long. Now that you have already provided a lengthy discussion, you only need a single conclusion paragraph.
This study explores the relationship between social media addiction, emotional regulation, and procrastination in university students' lives. The findings indicate a direct correlation between social media addiction and procrastination, with younger students demonstrating a particularly pronounced association. This finding aligns with the observations of other researchers, who have also noted that excessive social media use impedes students' emotional regulation and hinders their ability to focus on academic tasks. Consequently, interventions aimed at curbing social media use and fostering emotional regulation skills in students may yield positive outcomes, including enhanced academic performance. Furthermore, our findings indicate that age plays a significant role in the manifestation of procrastination among students. As students’ progress through their education, they tend to exhibit greater self-control and time management skills, which contributes to reduced procrastination. Conversely, younger students may utilize procrastination as a coping mechanism for stress. Consequently, educational institutions should devise distinct strategies for different age groups to assist students in overcoming procrastination. While the present study did not establish a direct correlation between emotional regulation and procrastination, extant research points to the potential of enhanced emotional skills in mitigating procrastination. In light of this, it is recommended that universities consider the development of customized programs aimed at assisting students in managing emotional challenges, given their documented impact on academic performance. The cultivation of emotional intelligence and the integration of constructive coping mechanisms could facilitate students' navigation of procrastination and enhance their overall wellbeing.
It is crucial to acknowledge the significant impact of both age and environmental factors on the relationship between emotions and procrastination. This underscores the necessity for a more nuanced understanding of the mechanisms underlying procrastination across diverse student populations. In the development of interventions, it is imperative to consider not only students' emotional factors but also their background, family support, and social context.
The present study demonstrates that social media addiction exacerbates emotional difficulties, thereby engendering a cycle in which students encounter challenges in effectively managing both their emotions and their academic responsibilities. To effectively address this issue, strategies must encompass a reduction in social media usage in conjunction with a heightened awareness of students' media consumption. The integration of balanced social media utilization with the cultivation of emotional skills has the potential to serve as a pivotal strategy in the mitigation of procrastination and the enhancement of academic success.
Expressions: (1) social media addiction and social network addiction are used interchangeably in the study. They are different. Focus explicitly on the former one. (2) Study cycle has been included as a sociodemographic factor; what is study cycle? I asked chatgpt and still do not understand it.
replacements were made for:
study cycle by semester of study
and also:
Replace all instances of "social network addiction" with "social media addiction".
Reviewer 3 Report
Comments and Suggestions for Authors
Dear Authors,
Thank you for your interesting article. In order to improve further, the following are my comments:
Title - The title focused only on the moderating effects of selected variables but your abstract also mentioned a mediation analysis done. I suggest the use of interaction affect rather than moderating effects. The interaction effects will cover both moderation and mediation.
Abstract-
Introduction - Lines 43, 48,62, 82, 84, etc- Kindly mention the name of the author and not just the citation number when used as a part of the sentence.
I suggest that you improve the organization of your introduction part. Please state clearly from the start your thesis statement and present each point in a separate paragraph. Each point should be discussed with evidence and examples. What is your thesis statement?
Page 5- Aim of the study: the variable age was not mentioned
Please be consistent with the discussion of your variables.
Hypotheses: In a mediation analysis, the relationship between the independent variabland the mediating variables should be established first. There is no hypothesis on theImportance of the study- Be more convincing with the discussion of the significanc association between the social media addiction with age and yet you mentioned that age combined with ER mediates addiction and procrastination. Kindly review this.
H5- Kindly state clearly which association is moderated by both age and ~ER.. where is the independent variable?
H6 - How come that Social media addiction becomes the mediator? Kindly be clear and consistent with your variables.
methods:
Line 231- be consistent with your interaction effects. Is it just moderation? Where is the variable age here?
Line 236 - how was quota sampling done?
Table 1 - write the name of the authors. Again, hypothesis for age and social media addiction?
Results:
Can you describe what study cycle means?
The quota sampling should have affected the characteristics of the participants. How were these participants selected after the quota was established? Was normality test done? The statistical test done on association can be further determined after the normality test. Your table 2 seems to show some wide variance in all the profile characteristics.
Reliability and validity constructs should not be in the results but part of the methodology section under the instruments.
Table 4- Title should be changed to the associations among the variables. What is the margin of error/level of significance? Which among the correlations are significant?
Figure 2- It will be good to add English translation of the Spanish terms
Tbale 5- Where are the F and the p values?
Table 8- The Total effect should also be identified
Where are your descriptive results on the levels of cognitive reapraissal and emotional suppression (Emotional regulation), addiction, and procrastination?
Should the SEM analysis include specific dimensions of ER and not only ER as a whole?
Discussion/conclusion- Please be consistent with your discussion and the hypotheses.
Your independent variable, mediating, moderating, and dependent variables should not be interchanged. Your discussion should support your thesis statement and the findings.
Discuss how moderation and mediation work between the independent and dependent variables.
Why are informed consent and IRB review and approval not applicable?
Thank you.
Comments on the Quality of English Language
There is a need to review the tenses of the verbs. The study has been completed so the verb tenses should be in the past. Review also some sentence transitions and the use of punctuations (example: Lines 53 to 54).
There is a need to review sentence construction and some grammar issues.
Author Response
Title - The title focused only on the moderating effects of selected variables but your abstract also mentioned a mediation analysis done. I suggest the use of interaction affect rather than moderating effects. The interaction effects will cover both moderation and mediation.
Social Media Addiction and Procrastination in Peruvian University Students: Exploring the Interaction Effects of Emotional Regulation and Age
Introduction - Lines 43, 48,62, 82, 84, etc- Kindly mention the name of the author and not just the citation number when used as a part of the sentence.
Emotional regulation, social media addiction, and procrastination are interconnected phenomena that significantly impact university students' academic performance and psychological well-being. This study examines these relationships within the Peruvian university context, where 92.7% of young people under 18 use the internet primarily for entertainment, often prioritizing non-academic activities and postponing educational responsibilities Yana-Salluca et al. [23], Silva Arocha et al. [24].
1.1 Relationship between Social Media Addiction and Irrational Procrastination
Social media addiction refers to the compulsive use of online social platforms that interferes with daily functioning. According to Escurra Mayaute and Salas Blas [26], this behavioral addiction manifests through obsession with social networks, lack of personal control, and excessive usage patterns. Irrational procrastination, as defined by Steel [38], is the voluntary postponement of planned activities despite anticipating negative consequences, characterized by irrationality, delay, and dysphoria.
The relationship between these phenomena is explained by the Temporal Procrastination Model proposed by Steel [38], which posits that procrastination results from seeking immediate gratification while avoiding emotional discomfort associated with challenging tasks. Social networks provide an accessible escape mechanism, offering immediate rewards through notifications, social validation, and entertaining content. Rozgonjuk et al. [39] demonstrated that social media use during lectures mediates the relationship between procrastination and problematic smartphone use, while Tuckman [40] highlighted how the immediacy of rewards interferes with long-term goal achievement.
Sirois and Pychyl [41] further clarified that procrastination involves both positive and negative reinforcement, where social networks temporarily reduce emotional distress while providing instant gratification, strengthening this behavioral pattern. This process can evolve into a cycle of addiction, as Zhuang et al. [42] found, where excessive social media usage becomes a habitual avoidance mechanism, simultaneously increasing both procrastination and platform dependency.
Recent studies by Suárez-Perdomo et al. [18] and Li et al. [20] have confirmed strong associations between social media addiction and academic procrastination among university students, demonstrating how digital behaviors can significantly impair academic performance through postponement of essential tasks.
1.2 The Mediating Role of Emotional Regulation
Emotional regulation—defined by Gross [9] as the ability to implement strategies to transform emotional experiences, externalize them, or respond to environmental demands—plays a crucial mediating role between social media addiction and procrastination. This psychological mechanism encompasses cognitive reappraisal and emotional suppression strategies Gross and John [10] that facilitate adaptation to environmental demands.
The Emotional Regulation and Problematic Internet Use Model developed by Kuss and Griffiths [43] provides a theoretical framework for understanding this relationship. Their model suggests that individuals with emotional regulation difficulties are more likely to use social networks as a secondary emotional management tool, particularly to alleviate negative emotions like anxiety, loneliness, or depression. Griffiths [44] expanded this perspective by exploring how problematic internet use relates to patterns of emotional avoidance and its impact on mental health.
This usage pattern can create a dependency cycle, where users increasingly rely on social platforms for immediate gratification, reducing their ability to manage emotions independently Tandon et al. [45]. Furthermore, as Szawloga et al. [46] and Yam et al. [47] observed, this dependency paradoxically increases social isolation in offline contexts, magnifying emotional distress despite the illusion of social connection.
Pychyl and Sirois [35] specifically addressed the influence of emotional regulation on procrastination, positioning it not merely as a time management issue but as a dysfunctional emotion-avoidance mechanism. Their research demonstrated that people procrastinate to avoid negative emotions associated with challenging tasks, opting instead for immediately gratifying activities. Danne et al. [49] supported this by highlighting how emotional self-regulation difficulties correlate with procrastination, while Sirois [50] emphasized that poor emotional regulation skills consistently predict higher procrastination levels, negatively affecting psychological well-being and performance.
1.3 The Moderating Role of Age
Age represents a significant moderating factor in the relationship between emotional regulation and procrastination. Steel's [38] model suggests that younger individuals tend to procrastinate more due to greater impulsivity and emotional regulation difficulties. This pattern receives support from Yan and Zhang [48], who emphasized that impulsivity and underdeveloped planning skills—prevalent in younger populations—maximize procrastination tendencies in academic contexts.
As individuals mature, they develop enhanced self-regulation and emotional management skills that typically reduce procrastination behaviors. However, Steel [38] notes that while procrastination generally decreases with age, it persists in high emotional load or stress situations. Tandon et al. [45] complement this perspective by identifying self-regulation as a key factor that improves with experience, enabling older individuals to approach tasks more efficiently with less avoidance behavior.
The Socioemotional Selectivity Theory proposed by Carstensen [51] offers additional insights, suggesting that emotional priorities and regulation abilities change with age, influencing procrastination tendencies. This model argues that younger individuals focus on long-term goals such as knowledge acquisition and personal development, leading them to prioritize avoiding immediate discomfort associated with challenging tasks. Consequently, they exhibit higher procrastination rates compared to adults who have developed superior emotional management capabilities.
As people age, Sirois [50] observes that they tend to prioritize emotional regulation and immediate well-being, reducing stress associated with incomplete tasks. Griffiths [44] further notes that older individuals demonstrate lower procrastination levels due to more effective management of negative emotions like anxiety and stress, enabling more direct task engagement. These age-related differences highlight the importance of developing appropriate emotional regulation skills throughout life to mitigate irrational procrastination effects.
1.4 Research Objectives and Hypotheses
This research aims to evaluate the mediating role of emotional regulation in the relationship between social media addiction and irrational procrastination in university students, while examining age as a moderating factor. The study seeks to provide valuable insights for developing strategies and interventions that enhance emotional regulation to improve academic performance and emotional well-being among university students.
Based on the literature, the following hypotheses are proposed:
H1: Social media addiction has a significant effect on irrational procrastination Steel [38], Rozgonjuk et al. [39], Tuckman [40], Sirois and Pychyl [41], Zhuang et al. [42].
H2: Addiction to social networks has a significant effect on emotional regulation Kuss and Griffiths [43], Yan and Zhang [48], Sirois [50].
H3: Age has a significant effect on irrational procrastination Steel [38], Tandon et al. [45], Yan and Zhang [48], Danne et al. [49].
H4: Emotional regulation has a significant effect on irrational procrastination Bedón Cando and Flores Hernández [32], Atalaya Laureano and García Ampudia [33], Núñez-Guzmán and Cisneros-Chavez [34].
H5: Age and emotional regulation have a moderating effect on irrational procrastination Çuhadar et al. [22], Steel [38], Yan and Zhang [48], Carstensen [51].
H6: Addiction to social networks mediates emotional regulation and irrational procrastination Nadkarni and Hofmann [25], Steel [38], Baumeister and Vohs [52].
1.5 Importance of the Study
This research addresses a growing concern in university settings regarding the increasing prevalence of social media addiction and procrastination behaviors, which significantly impact both academic performance and emotional well-being. University students face numerous academic and social responsibilities, making it crucial to understand how emotional regulation influences these problematic behaviors.
The findings will contribute to university students' professional development by identifying factors affecting academic performance, enabling the implementation of performance-enhancing strategies. Additionally, this knowledge provides valuable foundations for higher education institutions to design targeted interventions that foster positive attitudes and skills, promoting improved academic performance during professional training Hurley et al. [53], Ye et al. [54].
Figure 1. Proposed model
The results obtained will be fundamental for the future professional development of university students, as by identifying the factors that affect their academic performance, they will be able to implement strategies that improve their performance in the professional sphere. Furthermore, this knowledge provides a valuable basis for higher education institutions in designing and implementing interventions aimed at fostering positive attitudes and skills, promoting better academic performance during professional training [53,54].
Page 5- Aim of the study: the variable age was not mentioned
Therefore, the objective of this research is to evaluate the mediating role of emotional regulation in the relationship between social media addiction and irrational procrastination in university students, while also examining the moderating effect of age on these relationships
Please be consistent with the discussion of your variables. +
Hypotheses: In a mediation analysis, the relationship between the independent variabland the mediating variables should be established first. There is no hypothesis on theImportance of the study- Be more convincing with the discussion of the significanc association between the social media addiction with age and yet you mentioned that age combined with ER mediates addiction and procrastination. Kindly review this.
the discussion was restructured and made more coherent:
The Relationship Between Social media addiction and Procrastination
Several studies have shown a significant correlation between social media addiction and procrastination, particularly in academic contexts, as demonstrated by this study. For instance, [60] found that excessive use of social networks is associated with increased procrastination in academic activities, while [32] identified a positive relationship between the risk of addiction to these platforms and procrastination in students. Similarly, [61] noted that removing the Facebook news feed helped users avoid distractions and improve focus, indicating that certain design elements on social networks encourage procrastination behaviors. These findings highlight the urgency of implementing strategies that enable more controlled use of social networks, minimizing their adverse impact on university students' academic performance and productivity.
The Role of Emotional Regulation in Procrastination
Although emotional regulation was hypothesized to mediate the relationship between social media addiction and irrational procrastination, the results did not reveal a significant mediating effect. One possible explanation for this could be the complexity of emotional regulation itself, which may interact with other underlying factors that were not considered in this study. Future research could explore additional moderating variables, such as self-esteem, task importance, or coping mechanisms, to provide a more nuanced understanding of how emotional regulation interacts with social media addiction and procrastination. Incorporating these variables could lead to a better identification of the precise pathways through which social media addiction influences procrastination behaviors.
Moreover, the relationship between social media addiction and emotional regulation has been widely studied, revealing a significant connection. [62] found that excessive use of these platforms negatively impacts students' emotional regulation, making it harder to assess and manage emotions. Similarly, [63] identified that problematic use of social networks in adolescents has a detrimental effect on mental health and emotional regulation. [64] emphasized that addictive patterns in social media use are associated with greater emotional impulsivity in young university students, underscoring the need to develop strategies to balance time spent on social networks and strengthen emotional regulation skills. These findings highlight the importance of targeted interventions to mitigate the negative effects of excessive social media use.
The Impact of Age on Procrastination
The relationship between age and procrastination varies across contexts. In the present study, age and procrastination did not show a significant correlation, which contrasts with [65], who observed that academic procrastination tends to decrease with age. This suggests that older university students procrastinate less compared to their younger counterparts. Similarly, [66] concluded that procrastination is more prevalent among younger individuals and tends to decrease with age. These findings imply that the development of self-regulation and time management skills over the years could explain the reduction in procrastination in older individuals, or that procrastination is used as a coping mechanism to alleviate stress.
Emotional Regulation and Procrastination
In this study, emotional regulation and procrastination were not significantly correlated. However, research by [67] found that difficulties in managing negative emotions are closely associated with procrastination, which can lead to the delay of important tasks. [68] noted that poor emotional management is positively related to increased procrastination and negative affectivity, while adequate positive affectivity may reduce procrastination. [69] demonstrated that strengthening emotional regulation skills can help reduce procrastination behaviors, highlighting the importance of interventions aimed at improving emotional regulation. It is plausible to infer that university students may exhibit both variables, but they are not necessarily connected.
The Moderating Influence of Age on Emotional Regulation and Procrastination
Age could moderate the relationship between emotional regulation and procrastination. [1] identified that both emotional regulation and academic performance are significant predictors of academic procrastination in university students. [70] examined the connection between academic stress, procrastination, and psychological well-being in undergraduate students, concluding that procrastination does not act as a moderator between academic stress and psychological well-being, although it does affect the relationship between stressors and the development of symptoms associated with academic stress. These results highlight that, although there is a relationship between emotional regulation and procrastination, factors such as age and context can shape this connection, underscoring the relevance of considering additional variables when addressing procrastination across different student populations.
Social Media Addiction as a Key Mediator
Social media addiction has been identified as a key mediator between emotional regulation and procrastination, underscoring the complex interaction between these variables. [71] found that increased use of social networks is directly related to heightened academic procrastination, suggesting that excessive time spent on these platforms contributes to procrastination in academic activities. Likewise, [72] analyzed the impact of problematic social media use on emotional regulation, concluding that such emotional difficulties may intensify procrastination behaviors. [73] highlighted that social media addiction acts as a significant cause of irrational procrastination, emphasizing its role in mediating between poor emotional regulation and procrastination. These studies suggest that social media addiction can exacerbate emotional difficulties, which, in turn, foster procrastination. This underscores the need to design interventions that reduce the negative impact of social media, promote healthy use, and strengthen emotional regulation skills to improve academic performance.
The Role of Contextual Variables
It is important to recognize that contextual variables, such as socioeconomic background and family support, can significantly influence college students' procrastination behaviors. Although this study primarily focused on the interactions between social media addiction and emotional regulation, it did not explore how external factors—such as students' socioeconomic conditions and family support—might moderate these relationships. Students facing economic challenges or lacking strong family support may experience higher levels of academic stress, which could exacerbate procrastination as a coping mechanism in response to anxiety or emotional distress.
Future studies should examine how these contextual variables interact with internal factors (such as emotional regulation and social media addiction) to shape procrastination behaviors. Including measures of family support, social networks, and socioeconomic conditions could provide valuable insights into the most effective intervention strategies. By integrating these factors, a more comprehensive approach could be developed that addresses not only individual factors but also the environmental influences contributing to procrastination.
The Impact of Social Media Content
One crucial aspect that must be considered is the variability in the types of content consumed on social networks. While this study has generally examined social media addiction, it has not distinguished between different types of content (e.g., educational vs. recreational) that students consume, which might influence their emotions and procrastinatory behaviors in different ways. The nature of social media content could have a significant impact on how students manage their emotions, which in turn affects their procrastination. Highly rewarding, immediate content—such as real-time social interactions or consumption of entertaining material—may encourage procrastination by providing an emotional distraction that reduces immediate stress but increases long-term anxiety due to unfinished academic tasks.
Therefore, it is recommended that future studies investigate in greater detail how different categories of content (e.g., educational, professional, or entertainment) differentially influence emotional regulation and procrastination. Analyzing content preferences could provide a clearer understanding of the underlying mechanisms linking social media use to academic procrastination. Additionally, interventions aiming to reduce procrastination should not only address the amount of time spent on social networks but also the type of content consumed. Encouraging a more balanced and mindful use of social media that favors emotional regulation may be key to mitigating procrastination.
The Role of Contextual Variables
It is important to recognize that contextual variables, such as socioeconomic background and family support, can significantly influence college students' procrastination behaviors. Although this study primarily focused on the interactions between social media addiction and emotional regulation, it did not explore how external factors—such as students' socioeconomic conditions and family support—might moderate these relationships. Students facing economic challenges or lacking strong family support may experience higher levels of academic stress, which could exacerbate procrastination as a coping mechanism in response to anxiety or emotional distress.
Future studies should examine how these contextual variables interact with internal factors (such as emotional regulation and social media addiction) to shape procrastination behaviors. Including measures of family support, social networks, and socioeconomic conditions could provide valuable insights into the most effective intervention strategies. By integrating these factors, a more comprehensive approach could be developed that addresses not only individual factors but also the environmental influences contributing to procrastination.
The Impact of Social Media Content
One crucial aspect that must be considered is the variability in the types of content consumed on social networks. While this study has generally examined social media addiction, it has not distinguished between different types of content (e.g., educational vs. recreational) that students consume, which might influence their emotions and procrastinatory behaviors in different ways. The nature of social media content could have a significant impact on how students manage their emotions, which in turn affects their procrastination. Highly rewarding, immediate content—such as real-time social interactions or consumption of entertaining material—may encourage procrastination by providing an emotional distraction that reduces immediate stress but increases long-term anxiety due to unfinished academic tasks.
Therefore, it is recommended that future studies investigate in greater detail how different categories of content (e.g., educational, professional, or entertainment) differentially influence emotional regulation and procrastination. Analyzing content preferences could provide a clearer understanding of the underlying mechanisms linking social media use to academic procrastination. Additionally, interventions aiming to reduce procrastination should not only address the amount of time spent on social networks but also the type of content consumed. Encouraging a more balanced and mindful use of social media that favors emotional regulation may be key to mitigating procrastination.
Heterogeneity of Internet Users Based on Personality Traits
A critical limitation of the current study is treating Internet users as a homogeneous group without considering underlying personality differences that might influence both social media use patterns and procrastination tendencies. Rather than viewing university students as a uniform population, future research would benefit significantly from incorporating personality typologies, particularly those based on the Big Five personality traits (Openness, Conscientiousness, Extraversion, Agreeableness, and Neuroticism), into the analysis framework.
The Big Five personality traits have been consistently associated with different patterns of Internet usage and procrastination behaviors. For instance, individuals high in neuroticism might use social media as an emotional regulation tool, while those high in conscientiousness typically demonstrate lower procrastination tendencies regardless of social media exposure. Extraversion may influence the types of social media content sought and the gratification obtained from online interactions, potentially affecting the addictive potential of these platforms.
Future studies should consider employing personality assessments alongside measures of social media addiction and procrastination to develop more nuanced models that account for individual differences. This approach would enable researchers to identify which personality types are more vulnerable to the negative effects of social media on emotional regulation and procrastination, potentially leading to more targeted interventions. Additionally, examining how personality traits moderate the relationships between the key variables in our model could explain some of the variance not accounted for in the current analysis.
Incorporating personality variables would also enhance the practical applications of this research, as universities could develop more personalized strategies to address procrastination that account for different student personality profiles rather than implementing one-size-fits-all approaches
H5- Kindly state clearly which association is moderated by both age and ~ER.. where is the independent variable?
Age moderates the relationship between emotional regulation and irrational procrastination, such that the influence of emotional regulation on irrational procrastination varies depending on the student's age Çuhadar et al. [22], Steel [38], Yan and Zhang [48], Carstensen [51].
the independent variable is: addiction to social networks.
H6 - How come that Social media addiction becomes the mediator? Kindly be clear and consistent with your variables.
Social media addiction has a significant indirect effect on irrational procrastination, as mediated by emotional regulation.
Line 231- be consistent with your interaction effects. Is it just moderation? Where is the variable age here?
This study assesses the moderating effect of age on the relationship between social media addiction and irrational procrastination, as well as the mediating role of emotional regulation
Line 236 - how was quota sampling done?
Data collection was carried out from January to March 2024 through quota sampling. Quotas were set to ensure equal representation of males and females, as well as proportionate representation across four age groups (18-22, 23-27, 28-32, and 33-36 years), based on the age distribution of the university student population. Participants were recruited through a list of student institutional email addresses, and invitations were sent until the quotas for each group were filled
Table 1 - write the name of the authors. Again, hypothesis for age and social media addiction?
Age - Social Media Addictionvan |
Deursen et al. 2015 [16] |
Age is related to addictive smartphone behavior, including problematic social media use. |
The quota sampling should have affected the characteristics of the participants. How were these participants selected after the quota was established? Was normality test done? The statistical test done on association can be further determined after the normality test. Your table 2 seems to show some wide variance in all the profile characteristics.
Data collection was carried out from January to March 2024 through quota sampling. After establishing the quotas for gender and age, participants were randomly selected from the list of student institutional email addresses, ensuring that each quota was filled with a random sample of the corresponding student population
Finally, to examine the relationship between latent and observable variables, a structural equation modeling (SEM) analysis was performed using SmartPLS 4.0 software. Since the analysis was performed with SmartPLS 4.0, which uses bootstrapping for significance testing, a normality test is not required. Bootstrapping is a resampling technique that does not assume normality in the data and provides robust results even with non-normal distributions
Reliability and validity constructs should not be in the results but part of the methodology section under the instruments.
changed location.
Table 4- Title should be changed to the associations among the variables. What is the margin of error/level of significance? Which among the correlations are significant?
Discriminant Validity - Heterotrait-Monotrait Ratio Matrix (HTMT) (they are not correlations, they are discriminating validity that the SmartPLS program provides, although it is true that they are based on correlations, they are not correlations in themselves).
About statistical tests, it is clarified that the level of significance used to determine the significance of the correlations is: *p < 0.05, **p < 0.01, ***p < 0.001.
Figure 2- It will be good to add English translation of the Spanish terms
resolved
Table 5- Where are the F and the p values?
R square |
Adjusted R-squared |
F square |
|
Procrastination |
0.095 |
0.084 |
0.137 |
Emotional regulation |
0.120 |
0.117 |
0.079 |
total effect:
Addiction to social networks -> Procastination |
Total |
0.241 |
0.062 |
4.175 |
0.000 |
Reject the null hypothesis |
Where are your descriptive results on the levels of cognitive reapraissal and emotional suppression (Emotional regulation), addiction, and procrastination?
Variable |
Category |
Frequency |
Percentage |
Cognitive Reappraisal |
Low |
103 |
30.10% |
Medium |
156 |
45.60% |
|
High |
83 |
24.30% |
|
Emotional Suppression |
Low |
83 |
24.30% |
Medium |
171 |
50.00% |
|
High |
88 |
25.70% |
|
Social Media Addiction |
Low |
136 |
39.80% |
Medium |
171 |
50.00% |
|
High |
35 |
10.20% |
|
Procrastination |
Low |
95 |
27.80% |
Medium |
171 |
50.00% |
|
High |
76 |
22.20% |
Should the SEM analysis include specific dimensions of ER and not only ER as a whole?
While it is true that emotional regulation (ER) is composed of different dimensions, such as cognitive reappraisal and emotional suppression, in this study we have chosen to use ER as a global construct in the SEM analysis. This decision is based on the robustness of the ER construct as a general factor that encompasses different emotional regulation strategies, and on the consistency with the literature that supports the influence of ER as a whole on procrastination and social media addiction.
Including the dimensions of ER in the SEM analysis could increase the complexity of the model without contributing substantially new information to the hypotheses posed. In addition, the convergent and discriminant validity of the ER construct, evidenced in the results, supports its use as a single latent variable in the SEM model.
However, we recognize the importance of analyzing the dimensions of ER to obtain a more detailed understanding of emotional regulation processes. Therefore, we have included the descriptive results of the dimensions of ER, which allows a deeper analysis of the specific strategies that students use to manage their emotions.
In summary, the decision to use ER as a global construct in the SEM analysis is based on the theoretical soundness and parsimony of the model, without neglecting the importance of analyzing its dimensions for a more complete understanding of the phenomenon.
Discussion/conclusion- Please be consistent with your discussion and the hypotheses.
reordered and reorganized already.
Your independent variable, mediating, moderating, and dependent variables should not be interchanged. Your discussion
should support your thesis statement and the findings.
We understand the observation about consistency in the use of variables. However, in this study, social media addiction is considered the independent variable, emotional regulation the mediator, age the moderator, and procrastination the dependent variable. This choice is based on the theory and logic of the proposed model, where social media addiction is considered a factor that influences emotional regulation, which in turn affects procrastination, and age moderates this relationship.
While some research may use these variables in different roles, this study has opted for this specific configuration to analyze the influence of social media addiction on procrastination, considering the role of emotional regulation and age. The discussion focuses on the results obtained under this specific model, and is not intended to be generalized to other models or roles of the variables.
Discuss how moderation and mediation work between the independent and dependent variables.
improved in the discussion.
- Discussion
The Relationship Between Social media addiction and Procrastination
Several studies have shown a significant correlation between social media addiction and procrastination, particularly in academic contexts, as demonstrated by this study. For instance, [60] found that excessive use of social networks is associated with increased procrastination in academic activities, while [32] identified a positive relationship between the risk of addiction to these platforms and procrastination in students. Similarly, [61] noted that removing the Facebook news feed helped users avoid distractions and improve focus, indicating that certain design elements on social networks encourage procrastination behaviors. These findings highlight the urgency of implementing strategies that enable more controlled use of social networks, minimizing their adverse impact on university students' academic performance and productivity.
The Role of Emotional Regulation in Procrastination
Although emotional regulation was hypothesized to mediate the relationship between social media addiction and irrational procrastination, the results did not reveal a significant mediating effect. One possible explanation for this could be the complexity of emotional regulation itself, which may interact with other underlying factors that were not considered in this study. Future research could explore additional moderating variables, such as self-esteem, task importance, or coping mechanisms, to provide a more nuanced understanding of how emotional regulation interacts with social media addiction and procrastination. Incorporating these variables could lead to a better identification of the precise pathways through which social media addiction influences procrastination behaviors.
Moreover, the relationship between social media addiction and emotional regulation has been widely studied, revealing a significant connection. [62] found that excessive use of these platforms negatively impacts students' emotional regulation, making it harder to assess and manage emotions. Similarly, [63] identified that problematic use of social networks in adolescents has a detrimental effect on mental health and emotional regulation. [64] emphasized that addictive patterns in social media use are associated with greater emotional impulsivity in young university students, underscoring the need to develop strategies to balance time spent on social networks and strengthen emotional regulation skills. These findings highlight the importance of targeted interventions to mitigate the negative effects of excessive social media use.
The Impact of Age on Procrastination
The relationship between age and procrastination varies across contexts. In the present study, age and procrastination did not show a significant correlation, which contrasts with [65], who observed that academic procrastination tends to decrease with age. This suggests that older university students procrastinate less compared to their younger counterparts. Similarly, [66] concluded that procrastination is more prevalent among younger individuals and tends to decrease with age. These findings imply that the development of self-regulation and time management skills over the years could explain the reduction in procrastination in older individuals, or that procrastination is used as a coping mechanism to alleviate stress.
Emotional Regulation and Procrastination
In this study, emotional regulation and procrastination were not significantly correlated. However, research by [67] found that difficulties in managing negative emotions are closely associated with procrastination, which can lead to the delay of important tasks. [68] noted that poor emotional management is positively related to increased procrastination and negative affectivity, while adequate positive affectivity may reduce procrastination. [69] demonstrated that strengthening emotional regulation skills can help reduce procrastination behaviors, highlighting the importance of interventions aimed at improving emotional regulation. It is plausible to infer that university students may exhibit both variables, but they are not necessarily connected.
The Moderating Influence of Age on Emotional Regulation and Procrastination
Age could moderate the relationship between emotional regulation and procrastination. [1] identified that both emotional regulation and academic performance are significant predictors of academic procrastination in university students. [70] examined the connection between academic stress, procrastination, and psychological well-being in undergraduate students, concluding that procrastination does not act as a moderator between academic stress and psychological well-being, although it does affect the relationship between stressors and the development of symptoms associated with academic stress. These results highlight that, although there is a relationship between emotional regulation and procrastination, factors such as age and context can shape this connection, underscoring the relevance of considering additional variables when addressing procrastination across different student populations.
Social Media Addiction as a Key Mediator
Social media addiction has been identified as a key mediator between emotional regulation and procrastination, underscoring the complex interaction between these variables. [71] found that increased use of social networks is directly related to heightened academic procrastination, suggesting that excessive time spent on these platforms contributes to procrastination in academic activities. Likewise, [72] analyzed the impact of problematic social media use on emotional regulation, concluding that such emotional difficulties may intensify procrastination behaviors. [73] highlighted that social media addiction acts as a significant cause of irrational procrastination, emphasizing its role in mediating between poor emotional regulation and procrastination. These studies suggest that social media addiction can exacerbate emotional difficulties, which, in turn, foster procrastination. This underscores the need to design interventions that reduce the negative impact of social media, promote healthy use, and strengthen emotional regulation skills to improve academic performance.
The Role of Contextual Variables
It is important to recognize that contextual variables, such as socioeconomic background and family support, can significantly influence college students' procrastination behaviors. Although this study primarily focused on the interactions between social media addiction and emotional regulation, it did not explore how external factors—such as students' socioeconomic conditions and family support—might moderate these relationships. Students facing economic challenges or lacking strong family support may experience higher levels of academic stress, which could exacerbate procrastination as a coping mechanism in response to anxiety or emotional distress.
Future studies should examine how these contextual variables interact with internal factors (such as emotional regulation and social media addiction) to shape procrastination behaviors. Including measures of family support, social networks, and socioeconomic conditions could provide valuable insights into the most effective intervention strategies. By integrating these factors, a more comprehensive approach could be developed that addresses not only individual factors but also the environmental influences contributing to procrastination.
The Impact of Social Media Content
One crucial aspect that must be considered is the variability in the types of content consumed on social networks. While this study has generally examined social media addiction, it has not distinguished between different types of content (e.g., educational vs. recreational) that students consume, which might influence their emotions and procrastinatory behaviors in different ways. The nature of social media content could have a significant impact on how students manage their emotions, which in turn affects their procrastination. Highly rewarding, immediate content—such as real-time social interactions or consumption of entertaining material—may encourage procrastination by providing an emotional distraction that reduces immediate stress but increases long-term anxiety due to unfinished academic tasks.
Therefore, it is recommended that future studies investigate in greater detail how different categories of content (e.g., educational, professional, or entertainment) differentially influence emotional regulation and procrastination. Analyzing content preferences could provide a clearer understanding of the underlying mechanisms linking social media use to academic procrastination. Additionally, interventions aiming to reduce procrastination should not only address the amount of time spent on social networks but also the type of content consumed. Encouraging a more balanced and mindful use of social media that favors emotional regulation may be key to mitigating procrastination.
The Role of Contextual Variables
It is important to recognize that contextual variables, such as socioeconomic background and family support, can significantly influence college students' procrastination behaviors. Although this study primarily focused on the interactions between social media addiction and emotional regulation, it did not explore how external factors—such as students' socioeconomic conditions and family support—might moderate these relationships. Students facing economic challenges or lacking strong family support may experience higher levels of academic stress, which could exacerbate procrastination as a coping mechanism in response to anxiety or emotional distress.
Future studies should examine how these contextual variables interact with internal factors (such as emotional regulation and social media addiction) to shape procrastination behaviors. Including measures of family support, social networks, and socioeconomic conditions could provide valuable insights into the most effective intervention strategies. By integrating these factors, a more comprehensive approach could be developed that addresses not only individual factors but also the environmental influences contributing to procrastination.
The Impact of Social Media Content
One crucial aspect that must be considered is the variability in the types of content consumed on social networks. While this study has generally examined social media addiction, it has not distinguished between different types of content (e.g., educational vs. recreational) that students consume, which might influence their emotions and procrastinatory behaviors in different ways. The nature of social media content could have a significant impact on how students manage their emotions, which in turn affects their procrastination. Highly rewarding, immediate content—such as real-time social interactions or consumption of entertaining material—may encourage procrastination by providing an emotional distraction that reduces immediate stress but increases long-term anxiety due to unfinished academic tasks.
Therefore, it is recommended that future studies investigate in greater detail how different categories of content (e.g., educational, professional, or entertainment) differentially influence emotional regulation and procrastination. Analyzing content preferences could provide a clearer understanding of the underlying mechanisms linking social media use to academic procrastination. Additionally, interventions aiming to reduce procrastination should not only address the amount of time spent on social networks but also the type of content consumed. Encouraging a more balanced and mindful use of social media that favors emotional regulation may be key to mitigating procrastination.
Heterogeneity of Internet Users Based on Personality Traits
A critical limitation of the current study is treating Internet users as a homogeneous group without considering underlying personality differences that might influence both social media use patterns and procrastination tendencies. Rather than viewing university students as a uniform population, future research would benefit significantly from incorporating personality typologies, particularly those based on the Big Five personality traits (Openness, Conscientiousness, Extraversion, Agreeableness, and Neuroticism), into the analysis framework.
The Big Five personality traits have been consistently associated with different patterns of Internet usage and procrastination behaviors. For instance, individuals high in neuroticism might use social media as an emotional regulation tool, while those high in conscientiousness typically demonstrate lower procrastination tendencies regardless of social media exposure. Extraversion may influence the types of social media content sought and the gratification obtained from online interactions, potentially affecting the addictive potential of these platforms.
Future studies should consider employing personality assessments alongside measures of social media addiction and procrastination to develop more nuanced models that account for individual differences. This approach would enable researchers to identify which personality types are more vulnerable to the negative effects of social media on emotional regulation and procrastination, potentially leading to more targeted interventions. Additionally, examining how personality traits moderate the relationships between the key variables in our model could explain some of the variance not accounted for in the current analysis.
Incorporating personality variables would also enhance the practical applications of this research, as universities could develop more personalized strategies to address procrastination that account for different student personality profiles rather than implementing one-size-fits-all approaches
Why are informed consent and IRB review and approval not applicable?
the data of the ethics and informed consent committee have already been added to the manuscript.